# FBXO32 promotes microenvironment underlying epithelial-mesenchymal transition via CtBP1 during tumour metastasis and brain development

Sanjeeb Kumar Sahu[1], Neha Tiwari[2], Abhijeet Pataskar[1], Yuan Zhuang[1], Marina Borisova[1], Mustafa Diken[3], Susanne Strand[4], Petra Beli[1] & Vijay K. Tiwari[1]

The set of events that convert adherent epithelial cells into migratory cells are collectively known as epithelial–mesenchymal transition (EMT). EMT is involved during development, for example, in triggering neural crest migration, and in pathogenesis such as metastasis. Here we discover FBXO32, an E3 ubiquitin ligase, to be critical for hallmark gene expression and phenotypic changes underlying EMT. Interestingly, FBXO32 directly ubiquitinates CtBP1, which is required for its stability and nuclear retention. This is essential for epigenetic remodeling and transcriptional induction of CtBP1 target genes, which create a suitable microenvironment for EMT progression. FBXO32 is also amplified in metastatic cancers and its depletion in a NSG mouse xenograft model inhibits tumor growth and metastasis. In addition, FBXO32 is essential for neuronal EMT during brain development. Together, these findings establish that FBXO32 acts as an upstream regulator of EMT by governing the gene expression program underlying this process during development and disease.

[1] Institute of Molecular Biology (IMB), Ackermannweg 4, 55128 Mainz, Germany. [2] Institute of Physiological Chemistry, University Medical Center, Johannes Gutenberg University, 55131 Mainz, Germany. [3] TRON - Translational Oncology at the University Medical Center of the Johannes Gutenberg University gGmbH, Freiligrathstrasse 12, 55131 Mainz, Germany. [4] Department of Internal Medicine I, University Medical Center, Johannes Gutenberg University, Obere Zahlbacher Straße 63, 55131 Mainz, Germany. Correspondence and requests for materials should be addressed to V.K.T. (email: v.tiwari@imb-mainz.de)

Epithelial cells remain in close contact with their neighbors and maintain an apical–basal axis of polarity by the sequential arrangement of adherent junctions, desmosomes, and tight junctions[1]. Following the induction of epithelial to mesenchymal transition (EMT) program, cells undergo molecular and phenotypic remodeling that involves changes like cytoskeletal reorganization and loss of cell–cell junctions. This allows epithelial cells to escape from their original location by acquiring a migratory, mesenchymal identity[2]. Such a dramatic change in cell fate is essential for key processes during embryonic development, such as embryo implantation, embryonic layer formation, gastrulation and neural tube formation. In adults, this change in cell

fate is important for processes like tissue regeneration and wound healing[1–4]. However, aberrant activation of the EMT program is associated with disease phenotypes such as organ fibrosis[5] and tumor metastasis[4, 6].

A number of signaling pathways, such as TGF-β, FGF, EGF, HGF, Wnt/β-catenin and Notch are known to induce EMT[1, 7]. Among these, TGF-β is known to be the most potent and a prototypic inducer of EMT in various contexts, including development and cancer metastasis, whereas the others have more context-specific functions[2, 8]. A large number of evidences have established that the microenvironment plays a critical role during initiation and progression of EMT[9]. However, the effectors, through which TGF-β mediates remodeling of the microenvironment to promote EMT, remain poorly explored. It is also well established that EMT relies on defined, genome-wide transcriptional reprogramming[10, 11]. A number of transcription factors are implicated in this process, including SNAIL, ZEB, and several basic helix-loop-helix proteins[8, 10, 12]. Importantly, these transcription factors modulate gene expression in partnership with co-regulator proteins such as CTBPs[13–15]. Several studies have reported that CtBPs form complexes with a variety of epigenetic regulators or corepressor complexes that recruit epigenetic regulators[13, 15, 16]. CtBPs are also known to undergo dynamic posttranslational modifications which influence their stability or subcellular localization[14]. Recent studies have suggested dispensability of the established key EMT transcription factors in driving metastasis and further vouched for a need of exploring more potent factors driving this process[17–19].

Advances in proteomics have begun to highlight the role of post-translational modifications during EMT and their enormous complexity and regulatory potential[8, 20]. Ubiquitination via the ubiquitin–proteasome system governs diverse cellular processes, such as cell proliferation, cell cycle progression, transcription and apoptosis[21]. F-box proteins, the substrate-recognition subunits of Skp1–Cullin1–F-box protein E3 ligase complex, play pivotal roles in ubiquitination and subsequent activation or degradation of target proteins, depending on the lysine residue of ubiquitin (Ub) that participates in the formation of poly-Ub chains[22, 23]. Recent studies have shed light on the biological functions attributed by these F-box proteins[24]. FBXO32 (also known as Mafbx/Atrogin1) was first identified as a muscle-specific F-box protein, and further studies indicated its importance during heart development and muscle homeostasis[25, 26]. This protein is also induced upon stress, for instance, during serum starvation and hypoxia, and it functions as an apoptosis regulator[27, 28]. In a previous study, *FBXO32* promoter was shown to be repressed by DNA methylation in ovarian cancer and its expression was correlated with shorter progression-free survival[29]. FBXO32 has been implicated in the regulation of transcription factor stability (e.g., Myc) and localization (e.g., Foxo1 and Foxo3a)[26, 30]. A very recent study showed an association between EMT and *Fbxo32* in tumors with acquired platinum resistance[31]. However, despite these advances and indications of its role in cancer, the function of FBXO32 in EMT progression, metastasis and its contributions to the gene regulatory circuitry underlying these processes remain completely unknown.

Here we discover FBXO32 to be required in various contexts of EMT, including in disease and development. We report that FBXO32 is substantially induced during EMT and plays a critical role in the transition towards mesenchymal identity by governing the required gene expression program. We show that FBXO32-dependent K63-linked ubiquitination of CtBP1 is required for its nuclear retention, which is essential for mediating transcriptional changes via epigenetic reprogramming of EMT target genes. These include various chemokines, chemokine receptors, and matrix metalloproteinases, thus promoting a suitable environment promoting EMT progression. FBXO32 is also highly amplified in a large panel of cancers, and its depletion severely impairs the metastatic properties of cancer cells both in vitro and in vivo. Furthermore, depletion of FBXO32 during brain development impairs neuronal EMT. Together, these observations establish FBXO32 among the most potent regulator of EMT by governing the gene expression program that underlies this process during both development and disease.

## Results

**FBXO32 is essential for epithelial–mesenchymal transition.** To identify so far unknown regulators of EMT, we employed an established model system in which normal murine mammary gland epithelial cells (NMuMG) undergo TGF-β-induced EMT with high synchrony and purity[10, 12] (Supplementary Fig. 1a, b). To examine global alterations in the transcription of coding and noncoding genes during EMT, we analyzed genome-wide transcription profiles obtained via RNA-seq from NMuMG cells treated with TGF-β at several consecutive time points representing untreated, early (day 1), intermediate (day 4), and late (day 7) stages of EMT[10]. To identify previously unknown regulators of EMT that function in the nuclear compartment, we performed a stepwise selection approach to obtain candidate genes that are differentially expressed during EMT, contain a nuclear localization signal, and are aberrantly expressed in various cancers (using the NextBio, cBioPortal, and Oncomine databases). We further analyzed the expression patterns of these candidates in other established model systems of EMT, including TGF-β-induced EMT in a mouse Ras-transformed breast epithelial cell line (EpRAS) and human primary mammary epithelial cells (HMECs) (Fig. 1a, Supplementary Fig. 1c, f). We retained only those candidates that showed consistent expression patterns. These candidates were finally tested for their functional role using loss-of-function experiments using a pool of four siRNAs targeting the same gene, followed by extensive phenotypic and molecular characterization. Among these candidates, *FBXO32* emerged as the most potent regulator of EMT.

**Fig. 1** FBXO32 is an essential regulator of phenotypic and transcriptional changes that are hallmark of EMT. **a** Schematic representation of in vitro EMT induction and the identification of previously unknown regulators. **b** Using RT-PCR, the levels of Fbxo32 mRNA in NMuMG cells during TGF-β-induced EMT were measured relative to Ctcf and plotted on the y-axis. **c** FBXO32 mRNA level was measured by RT-PCR in primary human breast epithelial cells (HMEC) undergoing TGF-β-induced EMT as described in Fig 1b. **d, e** Western blot analysis of FBXO32 in mouse NMuMG cell (**d**) and human HMECs (**e**) undergoing TGF-β-induced EMT. Lamin and β-Actin acted as a loading control. **f** Representative bright-field and immunofluorescence images showing the localization and expression levels of EMT marker proteins after 4 days (d4) of siRNA-mediated depletion of FBXO32 compared to non-targeting control (siControl) in NMuMG cells. Staining was performed to assess the expression with antibodies against the epithelial markers E-cadherin and ZO1, the mesenchymal marker N-cadherin, Fibronectin-1, Phalloidin (to visualize the actin cytoskeleton) and Paxillin (to detect focal adhesion plaques). Scale bar, 100 μm. **g, h** Using RT-PCR, the levels of FBXO32 and key EMT markers in NMuMG **g** and HMEC **h** cells transfected with either control siRNA or independent siRNAs against FBXO32 during TGF-β-induced EMT were measured relative to CTCF and plotted on the y-axis. All experiments were performed in biological triplicates unless otherwise specified. Error bars represent the SEM of three independent biological replicates ($n = 3$). *$p < 0.05$, **$p < 0.01$, ***$p < 0.001$, Student's t test. See also Supplementary Figs. 1–3

*Fbxo32* showed strong transcriptional induction during TGF-β-induced EMT in NMuMG cells (Fig 1b, d, Supplementary Fig. 2a). FBXO25, the closest evolutionarily conserved member to FBXO32 (based on sequence and protein domain similarity), remained transcriptionally unchanged during EMT (Supplementary Fig. 2b, c). Furthermore, *FBXO32* induction was dependent on TGF-β SMAD axis during onset of EMT (Supplementary Fig. 2d), in line with previous

observations[29, 32]. Moreover, among all of the F-box-containing proteins ($n = 69$), only *Fbxo32* showed significantly strong upregulation during EMT (Supplementary Fig. 2e). In line with these observations, a similar transcriptional induction of *Fbxo32* was observed in other EMT models in mouse (EpRas cells) as well as human (primary human mammary epithelial cells (HMEC) and MCF7 cells) (Fig. 1c, e, Supplementary Fig. 2f, g).

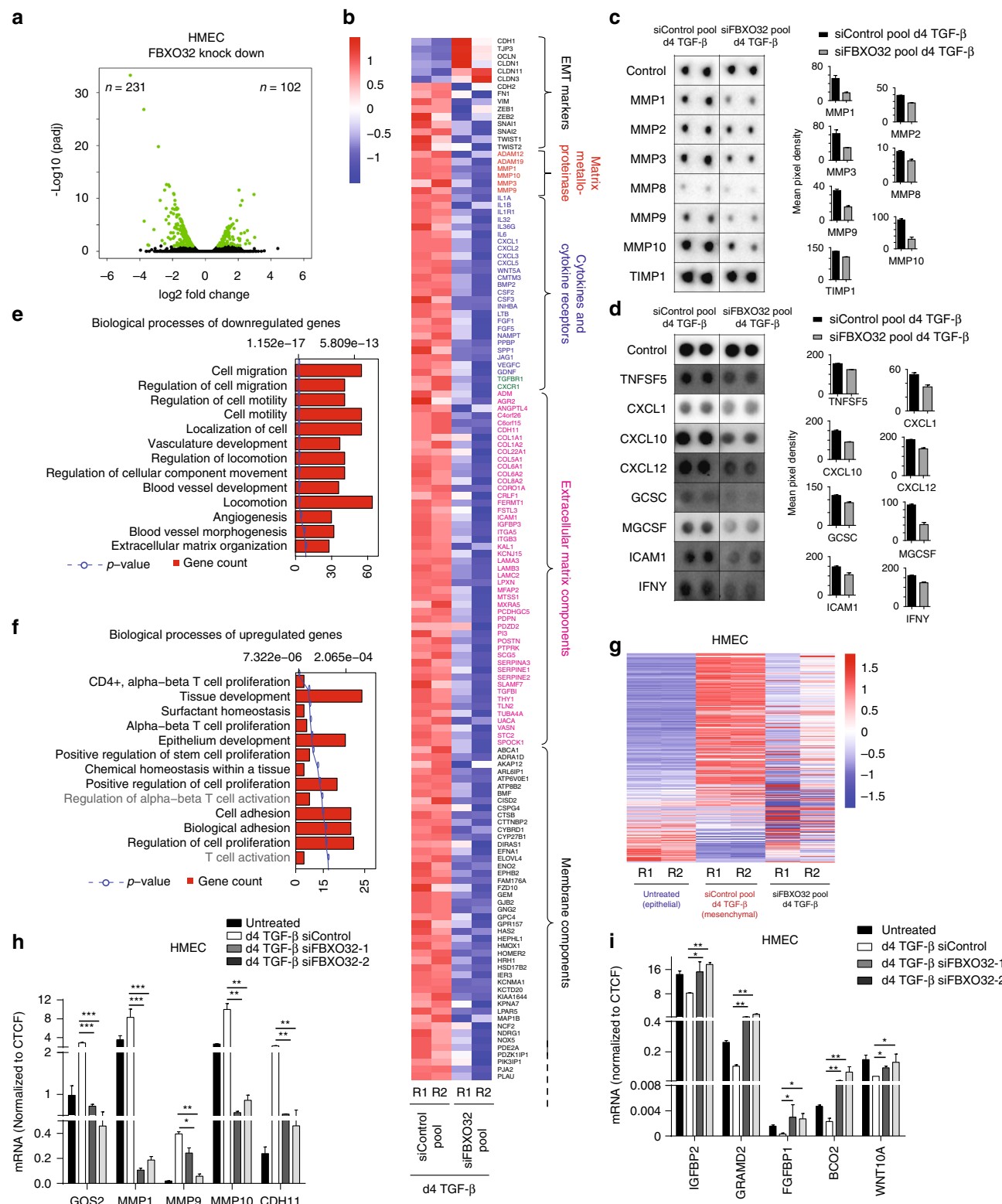

Next, we investigated the function of *Fbxo32* during TGF-β-induced EMT in mouse (NMuMG) cells using loss-of-function approaches both by independent siRNAs and a pool of siRNAs to ensure no off-target effects. Such depletion of Fbxo32 during EMT led to a strong blockage in acquiring a mesenchymal state in all tested cell types. This phenotype was confirmed by an immunofluorescence assay that showed noticeable retention in the epithelial markers (e.g., E-cadherin and ZO-1) at the membrane, inability to acquire proper levels of crucial mesenchymal markers (e.g., N-cadherin, Fibronectin), and failure to undergo cytoskeletal remodeling involving cortical actin, stress fibers, and focal adhesion formation (e.g., Phalloidin and Paxillin) (Fig. 1f). Such blockage in undergoing EMT upon *Fbxo32* depletion was further validated at RNA and protein levels for several key EMT markers (Fig. 1g, Supplementary Fig. 2h). In support of *Fbxo32*-dependancy of these changes, we could rescue the phenotypic and molecular alterations upon Fbxo32 depletion by overexpressing a siRNA-resistant *FBXO32* (Supplementary Fig. 2i, j). Importantly, all key findings were reproduced in established human EMT model systems ((TGF-β-induced EMT in HMEC and MCF7 cells)) (Fig. 1h, Supplementary Fig. 3a–d). Together, these observations reveal *FBXO32* is critical for phenotypic and transcriptional changes underlying EMT.

**FBXO32 induces genes that promote microenvironment underlying EMT**. Intrigued by the observed critical role of FBXO32 in phenotypic and molecular changes during EMT, we decided to investigate genome-wide transcriptional alterations following FBXO32 depletion by a pool of four siRNAs during EMT in HMEC. A computational analysis revealed that many genes were significantly differentially expressed, with more genes downregulated ($n = 231$) than upregulated ($n = 102$), suggesting a potential activating function of FBXO32 (Fig. 2a). This included several established hallmark EMT genes such as CDH1, TJP3, OCLN, CLDN1/3/11 (all epithelial makers), and CDH2, FN1, VIM, ZEB1/2, SNAI1/2, TWIST1/2, (all mesenchymal makers) that were upregulated and downregulated, respectively (Fig. 2b). Interestingly, a deeper investigation of the downregulated genes uncovered enrichment of important cytokines (interleukin and CXCL family members), cytokine receptors (CXCR and TGFBR1), extracellular matrix components (laminins and collagens) and matrix metalloproteinases (MMPs, e.g., MMP1/3/9/10), which are known to play critical roles in EMT (Fig. 2b). In line with these findings, an analysis of the secretome in these cells showed that the functional protein levels of these secretory components, including several MMPs and cytokines, were strongly downregulated upon FBXO32 depletion (Fig. 2c, d). The downregulated genes from RNA-seq experiments showed enrichment for ontologies related to cell migration, cell motility and extracellular matrix organization, while the upregulated genes showed enrichment for biological functions such as proliferation, epithelium development and cell adhesion (Fig. 2e, f).

Interestingly, when we analyzed the expression dynamics of these misregulated genes in epithelial cells, mesenchymal cells and cells depleted of FBXO32, we found that FBXO32 depleted cells retained a more epithelial-like gene expression profile and were unable to acquire a mesenchymal transcriptome (Fig. 2g). A number of these genes were additionally validated for their expression changes independently by RT-PCRs (Fig. 2h, i). These findings suggest that FBXO32 plays a critical role in EMT by regulating the underlying gene expression program, especially of secretory cytokine and cellular matrix associated factors that are essential to promote a suitable microenvironment for EMT progression.

**FBXO32 ubiquitinates CtBP1 to promote its nuclear retention**. Prompted by the critical role of FBXO32 in all the studied contexts of EMT, we sought to gain mechanistic insights into its function. While FBXO32 depletion resulted in transcriptome changes, given the known function of F-box-containing family members as a component of the Ub ligase complex, and with strong nuclear localization (Supplementary Fig. 4a, b), we hypothesized that its gene regulatory function likely involves regulating the activity of other transcriptional regulators via post-translational modifications. We therefore performed immuno-precipitation of FBXO32, followed by liquid chromatography-tandem mass spectrometry, to search for its possible interactors and substrates (Fig. 3a). Among the most enriched candidates detected by this analysis, we identified C-terminal-binding protein1 (CtBP1), which is known to play a key role in promoting EMT[15, 33, 34]. This interaction between FBXO32 and CtBP1 was further validated in independent co-immunoprecipitation assays both following their overexpression and at the endogenous level (Fig. 3b, c and Supplementary Fig. 4c, d). Intriguingly, this interaction was abrogated by deletion of the large F-box domain (115 aa, as predicted by InterPro, which encompasses the core F-box domain of 45 aa) (Supplementary Fig. 4d). Since CtBP1 is expressed at very similar levels during EMT and upon FBXO32 depletion (Supplementary Fig. 4e–g), we hypothesized that FBXO32 regulates its function via post-translational modifications.

FBXO32-mediated ubiquitination is known to cause both activation of signaling cascades and proteasome-mediated degradation[26, 30]. The overexpression of FBXO32 in human cells resulted in an increased ubiquitination of CtBP1, which can be blocked by Ub E1 inhibitor UBEI-41/ PYR-41[35] (Fig. 3d, Supplementary Fig. 4h). Moreover, FBXO32 overexpression led to an increase in CtBP1 protein levels, whereas FBXO32 depletion reduced the CtBP1 protein levels in both human and mouse EMT model systems (Fig. 3e and Supplementary Fig. 4i–k). Due to a poor transfection efficiency of primary HMEC cells, here we used HMLE cells that originate from HMEC cells and show a similar dynamics of EMT and FBXO32 induction (Supplementary Fig. 4l)[36, 37]. We investigated whether the FBXO32-dependent

---

**Fig. 2** FBXO32 induces genes critical for promoting a suitable microenvironment for EMT progression. **a** Volcano plots showing significantly differentially expressed genes upon FBXO32 depletion in HMECs undergoing EMT. **b** Heat map showing standardized expression (Z-scores) of key downregulated genes along with EMT markers, upon FBXO32 depletion in HMEC cells undergoing EMT for 4 days. **c, d** Representative immunoblot ($n = 3$) of an individual membrane array for human secretory MMPs (**c**) and cytokines (**d**) from HMEC cells upon FBXO32 depletion, and bar-graph showing their quantifications on the right side. Control wells act as loading controls in our analysis to allow comparison of different blots. **e, f** GO analysis of downregulated (**e**) and upregulated genes (**f**) upon FBXO32 knockdown in HMECs that underwent EMT for 4 days. **g** Heat map showing standardized expression (Z-scores) of differentially expressed genes upon FBXO32 depletion in Epithelial HMEC cells, HMEC that underwent EMT for 4 days and treated with control siRNA-pool or siRNA-pool against FBXO32. **h, i** The mRNA levels of downregulated genes (**h**) and upregulated genes (**i**) during EMT in control and FBXO32 knockdown HMECs that underwent EMT for 4 days were measured relative to CTCF via RT-PCR, and the results were plotted on the *y*-axis. Error bars represent the SEM of three independent biological replicates ($n = 3$). *$p < 0.05$, **$p < 0.01$, ***$p < 0.001$, Student's *t* test. See also Supplementary Figs. 2 and 3

ubiquitination of CtBP1 has consequences on the subcellular distribution of CtBP1. Strikingly, although CtBP1 is highly localized to the nucleus under normal cellular conditions, FBXO32 depletion resulted in CtBP1 localizing exclusively to the cytoplasm in both human and mouse cells (Fig 3f and Supplementary Fig. 5a). The degradation of proteins is known to be mediated via K48-linked poly-Ub chains while K63-linked

ubiquitination is associated with the activation of signaling pathways[38]. It is known that FBXO32 can perform both K48 and K63-linked ubiquitination, which results in proteosome-mediated degradation or nuclear localization of target proteins respectively[26, 30]. Our further analyses showed that FBXO32 mediates K63-linked ubiquitination, but not K48-linked ubiquitination of CtBP1 (Fig 3g, Supplementary Fig. 5b). Moreover previous

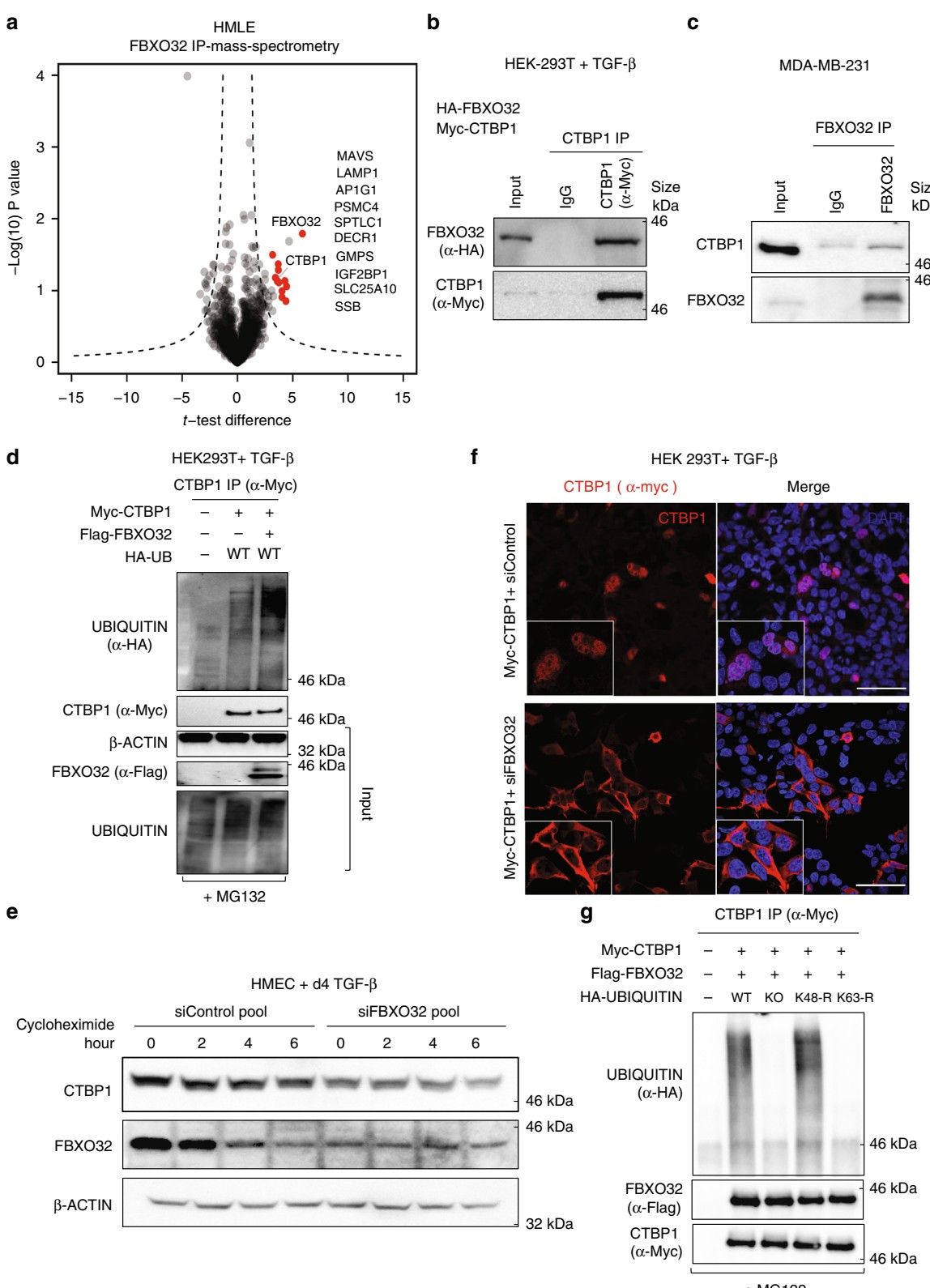

finding also indicates that additional non-K63 lysines are needed for full ubiquitination as it partially rescued the ubiquitination phenotype (Supplementary Fig. 5b). Together, these observations establish that FBXO32 interacts with and modifies CtBP1 to promote its stability and nuclear retention.

**Nuclear CtBP1 mediates epigenetic remodeling of EMT genes.** CtBP1 can repress or activate transcription by recruiting distinct classes of epigenetic regulators[39]. To substantiate our findings regarding a FBXO32-dependent role of CtBP1 in promoting EMT, we depleted CtBP1 during EMT in human primary breast epithelial cells. Such loss of CtBP1 led to an impaired EMT similar to that observed following FBXO32 depletion (Supplementary Fig. 6a). Encouraged by these findings, we performed genome-wide gene expression profiling (RNA-seq) following CtBP1 ablation in HMECs undergoing TGF-β-induced EMT. Similar to FBXO32 knockdown, CtBP1 depletion led to a larger number of downregulated genes compared to upregulated genes (Fig. 4a). Importantly, a comparative analysis of the genes downregulated upon CtBP1 depletion showed a highly significant overlap with the genes downregulated upon FBXO32 depletion (Fig. 4b). The overlapping genes included MMPs and Cytokines, which have established functions in signaling, cytoskeletal rearrangement and cell migration[40–43] (Fig. 4b, Supplementary Fig. 6b). Depletion of CtBP1 in FBXO32 overexpression background resulted in a reduced induction of its target genes (Supplementary Fig. 6c). These findings validated FBXO32 as an upstream regulator of CtBP1 during EMT.

Given our observations that FBXO32-mediated ubiquitination of CtBP1 is required for its stability and nuclear retention, we tested whether this is linked to CtBP1 chromatin binding and epigenetic regulatory function at its target genes. In line with our observations, we found that CtBP1 depletion impairs EMT (Supplementary Fig. 6a) and reduces expression of genes which were downregulated following FBXO32 knockdown (Fig. 4b, c). Importantly, loss of FBXO32 led to a significant reduction in CtBP1 occupancy at promoters of key EMT relevant genes (Fig. 4d). Since CtBP1 was previously reported to mediate gene regulation via epigenetic mechanisms[13, 16], we explored whether CtBP1 depletion influences chromatin state of its target loci. In line with the changes in gene expression, CtBP1 depletion led to a substantial decrease in chromatin accessibility at the majority of tested target promoters (Fig. 4e). Further analysis of active histone modifications H3K4me3 and H3K27ac at these CtBP1 target promoters revealed a significant decrease in their enrichment upon FBXO32 depletion (Fig. 4f and Supplementary Fig. 6d). Moreover, this also accompanied an increase in the repressive mark H3K9me3 at these sites (Fig. 4g). Interestingly,

CtBP1 knockdown also lead to similar epigenetic changes at the FBXO32 promoter, resulting in its transcriptional downregulation (Fig. 4c–g). This indicates a possible feed-forward loop between FBXO32 and CtBP1 that promotes EMT. All together, these findings led us to conclude that FBXO32 mediates transcriptional activation of key EMT genes, via CtBP1 targeting to their promoters, and subsequent epigenetic remodeling that generates a transcriptionally permissive state.

We investigated whether the genes regulated by FBXO32 and CtBP1 are enriched for specific transcription factor binding motifs. For this, we performed motif enrichment analysis at the promoter regions of the genes that were deregulated following the depletion of FBXO32 or CtBP1. This investigation at the promoter of downregulated genes upon FBXO32 depletion identified NFKB, activating transcription factor 3 (ATF3) and BATF as the top motifs, whereas similar analysis for downregulated genes upon CtBP1 knockdown identified E2F6, BATF and ATF3 as the top motifs (Fig. 4h, i). Since ATF3 was enriched in both FBXO32 and CtBP1 targets, we attempted to further validate these motif predictions using publically available genome-wide binding (ChIP-seq) data sets for ATF3 in human cells. Remarkably, majority of the FBXO32 and CtBP1 regulated gene loci showed experimental evidence of ATF3 occupancy (Fig. 4j and Supplementary Fig. 7a). Further analysis showed that while ATF3 pre-occupied these target sites in epithelial cells, the stability of its binding during EMT strongly relied on FBXO32 and CtBP1 (Fig. 4k, l). Moreover, ATF3 depletion during EMT led to a downregulation of these genes (Supplementary Fig. 7b). These findings imply that ATF3 may cooperate with CtBP1 in gene regulation at distinct loci during EMT.

Altogether, these range of findings conclude that the FBXO32-dependent nuclear localization of CtBP1 is essential for CtBP1 binding to its targets, and the epigenetic remodeling required for the gene expression changes, underlying EMT.

**Neuronal migration during brain development requires Fbxo32.** We next asked whether FBXO32 functions similarly during developmental EMT. In the developing brain during cortical development, radial glial cells in the ventricular zone (VZ) undergo asymmetric division and daughter cells migrate toward the cortical plate (CP), passing through the subventricular zone (SVZ) via an EMT-like mechanism[44] (Fig. 5a). Interestingly, analysis of laser micro-dissected samples from the VZ, SVZ and CP of the E14.5 mouse cortex revealed expression changes in several EMT genes that closely mimicked classical EMT (Fig. 5b). Further analysis of these data sets revealed a significant transcriptional induction of *Fbxo32* during neuronal EMT (Fig. 5c). These results were independently validated using

**Fig. 3** FBXO32-mediated K63-linked ubiquitination of CtBP1 is required for its stability and nuclear retention. **a** Analysis of FBXO32 interaction partners by affinity purification-mass spectrometry. Volcano plot of three replicate showing selected significantly enriched proteins in FBXO32 immunoprecipitated samples. HMLE cells were transfected with Flag-tagged FBXO32 or an empty vector and were induced with TGF-β for 4 days. FBXO32 and interaction partners were immunoprecipitated using Flag affinity resin, followed by their identification via LC-MS/MS. **b** Western blot of immunoprecipitated samples to validate FBXO32 and CtBP1 interactions. HEK293T cells were co-transfected with Myc-CtBP1 and Flag-HA-FBXO32 and anti-CtBP1 IP was performed. Immunoblot was performed to detect FBXO32 and CtBP1. **c** Western blot of immunoprecipitated samples to validate FBXO32 and CtBP1 endogenous interactions in MDA-MB-231 cell. FBXO32 IP was performed and Immunoblot was performed to detect CtBP1 and FBXO32. **d** HEK293 were transfected with various tagged construct as mentioned in the figure. Four hours before immunoprecipitation MG132 was added for all conditions. Western blot for CtBP1 (α-Myc) immunoprecipitated samples was performed to detect ubiquitin (α-HA) and CTBP1. β-Actin, FBXO32 and Ubiquitin were detected in Input samples as controls. **e** Western blot to access protein stability, showing CtBP1 and FBXO32 protein levels at various time points after cycloheximide treatment in HMEC cells transfected with control siRNA or siRNA against FBXO32. β-Actin acted as a loading control. **f** Immunofluorescence image showing CtBP1 localization in control and FBXO32-depleted HEK293T cells. Magnified images of the nuclei are provided in the lower left corner of each image. Scale bar, 100 μm. **g** A similar analysis as in **d**, but co-transfected with modified ubiquitin to demonstrate FBXO32-mediated K63-linked ubiquitination of CtBP1. Here, K63-R represents ubiquitin with lysine at position 63 modified to arginine. Similarly, in K48-R, lysine at position 63 modified to arginine. See also Supplementary Figs. 4–6

FAC-sorted CD133+ (VZ population) and NCAM+ cells (neuronal populations) (Fig. 5d). In addition, the transcriptional dynamics of *Fbxo32* was confirmed by RNA in situ hybridization datasets in the developing mouse cortex (Fig. 5e). Such *Fbxo32* upregulation during neuronal EMT in vivo also accompanied the induction of previously identified Fbxo32-dependent EMT-relevant genes (Supplementary Fig. 8).

To test the role of FBXO32 during cortical development, we performed in utero electroporation of mouse cortex[45] at E12.5 with GFP reporter plasmids containing either control shRNA or validated shRNA against Fbxo32 and the mice were euthanized at E16.5 for further analysis (Fig. 5f). While cells electroporated with control shRNA participated normally in neurogenesis and populated throughout the cortex four days post-electroporation,

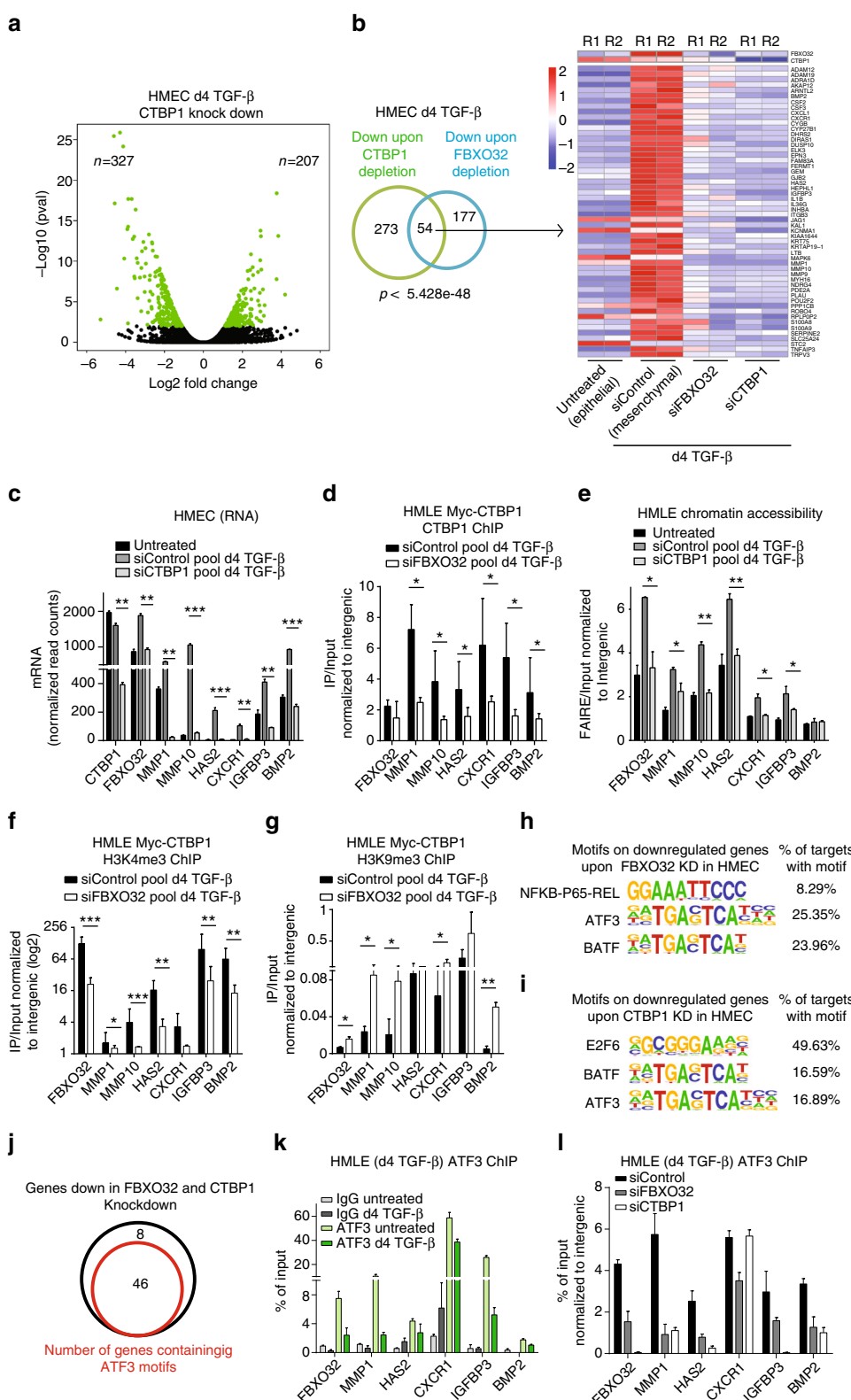

Fbxo32-depleted cells resided almost exclusively below the SATB2-positive layer, which marks neurons (Fig. 5g, h). These observations were further confirmed by a binning analysis of the electroporated cortices (Fig. 5g, i). Such retention of *Fbxo32*-depleted cells in the VZ/SVZ suggested a potential defect in cellular migration that is known to involve EMT. Moreover, FBXO32 was also found to be induced during human in vitro neurogenesis (Fig. 5j)[46]. Overall, these observations indicate FBXO32 as a critical player in neuronal EMT during brain development.

**Tumor aggressiveness correlates with FBXO32 expression levels.** Prompted by our observations of a critical function of FBXO32 during EMT, we investigated its expression in a wide range of human tumors. A large-scale analysis of several well-characterized human clinical expression data sets revealed an exceptionally strong amplification of *FBXO32* in a majority of human tumors (Fig. 6a). Computational analyses revealed significantly higher expression of *FBXO32* in various tumors compared to the matched normal tissues (Fig. 6b, c and Supplementary Fig. 9a–d). In support of our observations, *FBXO32* levels correlated positively with the levels of established mesenchymal markers and negatively with the levels of epithelial markers in different types of tumor examined (Supplementary Fig. 9e–f). A survival analysis of the clinical breast tumor data sets revealed a significant correlation between higher *FBXO32* expression and poor relapse-free and metastasis-free survival (Fig. 6d, e). Moreover, *FBXO32* expression showed the expected correlation with the levels of established EMT markers as well as the identified FBXO32-induced EMT genes in multiple cancers, corroborating its critical role during human tumor metastasis (Fig. 6f–i). A deeper analysis of various tumor data sets (breast, colon, gastric, lung, and ovarian tumor) further validated these findings (Supplementary Fig. 9g–l, and 10a–c). In addition, transcriptome analysis of a large set of cancer cell lines confirmed an enhanced expression of *FBXO32* in mesenchymal cancer cell lines as compared to epithelial counterparts (Fig. 6j). To ultimately validate these findings in clinical samples, we collected non-invasive and invasive breast tumor samples ($n = 20$ each) and analyzed the expression of *FBXO32* in these samples. In line with our previous observations, *FBXO32* expression was significantly higher in the invasive tumors compared to the non-invasive tumors (Fig. 6k). Overall, these comprehensive set of observations extend our findings to the clinic and suggest a role for *FBXO32* in promoting tumorigenicity and metastasis in humans.

**FBXO32 promotes tumorigenicity and metastasis in mouse model.** To functionally assess the contribution of FBXO32 to the mesenchymal state in a disease context, we investigated FBXO32 expression in certain tumor cell lines that are routinely used for such studies. In line with our earlier observations (Fig. 6j), FBXO32 expression was significantly higher in human mesenchymal cancer cell lines (MDA-MB-231 and BT547) compared to human epithelial cancer cell lines (MCF7 and MDA-MB-361) (Fig. 7a, b). Next, we depleted FBXO32 in these cell lines and assessed their proliferation and colony-forming capacity. While the epithelial cell lines did not exhibit any obvious effects, the mesenchymal cells showed significantly reduced proliferation (Supplementary Fig. 11a). Moreover, depletion of FBXO32 in the mesenchymal breast cancer cell line (MDA-MD-231) led to a significant reduction in their migration and invasion potential (Fig. 7c, d). Encouraged by these observations, we performed global gene expression profiling (RNA-seq) in FBXO32-depleted MDA-MB-231 cells. Similar to our previous findings during EMT in HMEC, FBXO32 depletion led to a significant downregulation of 704 genes and an upregulation of only 280 genes (Fig. 7e). The downregulated genes included many migration-relevant genes, such as cytokines (e.g., IL6/8/18, CXCL2/3) (Supplementary Fig. 11b).

MDA-MB-231 cells have been reported to form tumors and metastasize to distant organs in mouse models[47]. To further investigate the role of FBXO32 in primary tumor growth and metastasis formation, we generated MDA-MB-231 cells (containing GFP and a luciferase reporter) with stable integration of a control shRNA or validated shRNA against FBXO32. Our analysis showed no defects in proliferation or signs of apoptosis in these cells stably depleted of FBXO32 (Supplementary Fig. 11c–e). Similarly, in vitro sphere formation assay did not reveal any obvious alterations in cell proliferation and colony forming ability (Supplementary Fig. 11f, g). These cells (pool of cells with stable integration) were xenografted into the mammary fat pad of NOD SCID gamma (NSG) mice ($n = 6$), and tumor growth and metastasis to distant organs were quantified at regular intervals. The primary tumor growth was severely reduced in the absence of FBXO32 over time (Fig. 7f, g). Importantly, the FBXO32-depleted tumor cells were unable to migrate to distant organs, as measured by in vivo luciferase activity at day 50 (Fig. 7h–k). These data were verified by analyzing the parental tumor cells using GFP staining in the lungs, which is known to be the preferred destination of these tumor cells following metastasis (Fig. 7l). Gene expression analysis of primary tumors depleted of FBXO32 also validated downregulation of previously identified FBXO32-dependent genes (Fig. 7m). Hematoxylin and eosin

**Fig. 4** FBXO32-dependent nuclear localization of CtBP1 is essential for epigenetic and transcriptional remodeling of critical EMT genes. **a** Differential expression analysis showing significantly differentially expressed genes upon CtBP1 depletion in HMECs undergoing EMT for 4 days. **b** A venn diagram showing the overlap of downregulated genes upon CtBP1 and FBXO32 depletion in HMECs undergoing EMT for 4 days and on the right hand, heat map showing their expression along with epithelial HMEC. **c** RNA-seq data showing the mRNA levels of key EMT-associated deregulated genes upon CtBP1 depletion. The y-axis represents the normalized tag count. **d** ChIP assay using an anti-Myc antibody to assess the binding of Myc-CtBP1 to its target promoter in HMLE cells induced with TGF-β for 4 days and transfected with either control siRNA or siRNA against FBXO32. q-PCR was performed for the indicated gene promoters, and gene enrichment was plotted on the y-axis as the ratio of precipitated DNA (bound) to total input DNA and normalized with intergenic region. **e** FAIRE assay was performed to assess the chromatin accessibility of EMT-associated deregulated genes in HMLE cells induced with TGF-β for 4 days and transfected with control siRNA or siRNA against CtBP1. Quantitative PCR was performed for the indicated gene promoters, and gene enrichment was plotted on the y-axis as the ratio of precipitated DNA (bound) to total input DNA and normalized with intergenic region. **f, g** A similar analysis as in **d** but ChIP performed for the H3K4me3 mark (**f**) and H3K9me3 mark (**g**). **h, i** Enriched motifs at the promoters of downregulated genes upon FBXO32 and CtBP1 knockdown (**h**) and (**i**) respectively. **j** A Venn diagram showing the presence of ATF3 motifs on or in close proximity to the promoters of genes that were downregulated upon FBXO32 and CtBP1 knockdown in HMEC. **k** A similar analysis as in **d** but with ChIP performed in HMLE cell for ATF3 and gene enrichment was plotted on the y-axis as % of total input. IgG were run as a control. **l** ATF3 ChIP were performed in HMLE cells undergoing TGF-β induced EMT and depleted for FBXO32 or CtBP1 and gene enrichment was plotted on the y-axis as % of total input and normalized to intergenic region. For all above experiments the error bars represent the SEM of three independent biological replicates ($n = 3$). *$p < 0.05$, **$p < 0.01$, ***$p < 0.001$, Student's *t* test. See also Supplementary Fig. 7

(H&E) staining on these tumors showed a more compact organization of cells in FBXO32 depleted primary tumors as compared to the control tumors, which may be linked to the reduced metastatic behavior of FBXO32-depleted tumor cells (Fig. 7n). We also observed a clear decrease in the levels of secretory cytokines such as IL8, NGFR and GM-CSF in FBXO32-depleted tumors which may be associated with their compromised migratory potetial (Fig. 7o). Furhermore, MDA-MB-231 cells in culture as well as primary tumors from mouse xenograft experiments showed strongly reduced CtBP1 levels in the nuclear compartment following FBXO32 depletion (Supplementary Fig. 11h, i). Altogether these in vivo observations establish a

critical role for FBXO32 in promoting tumorigenicity and metastasis.

## Discussion

The differentiation of epithelial cells into motile mesenchymal cells, a process known as EMT, is integral in development and wound healing and it contributes pathologically to fibrosis and cancer progression. EMT involves action of transcription factors such as SNAIL, ZEB, and TWIST, the functions of which are finely regulated at the transcriptional, translational and post-translational levels. These key transcription factors utilize various

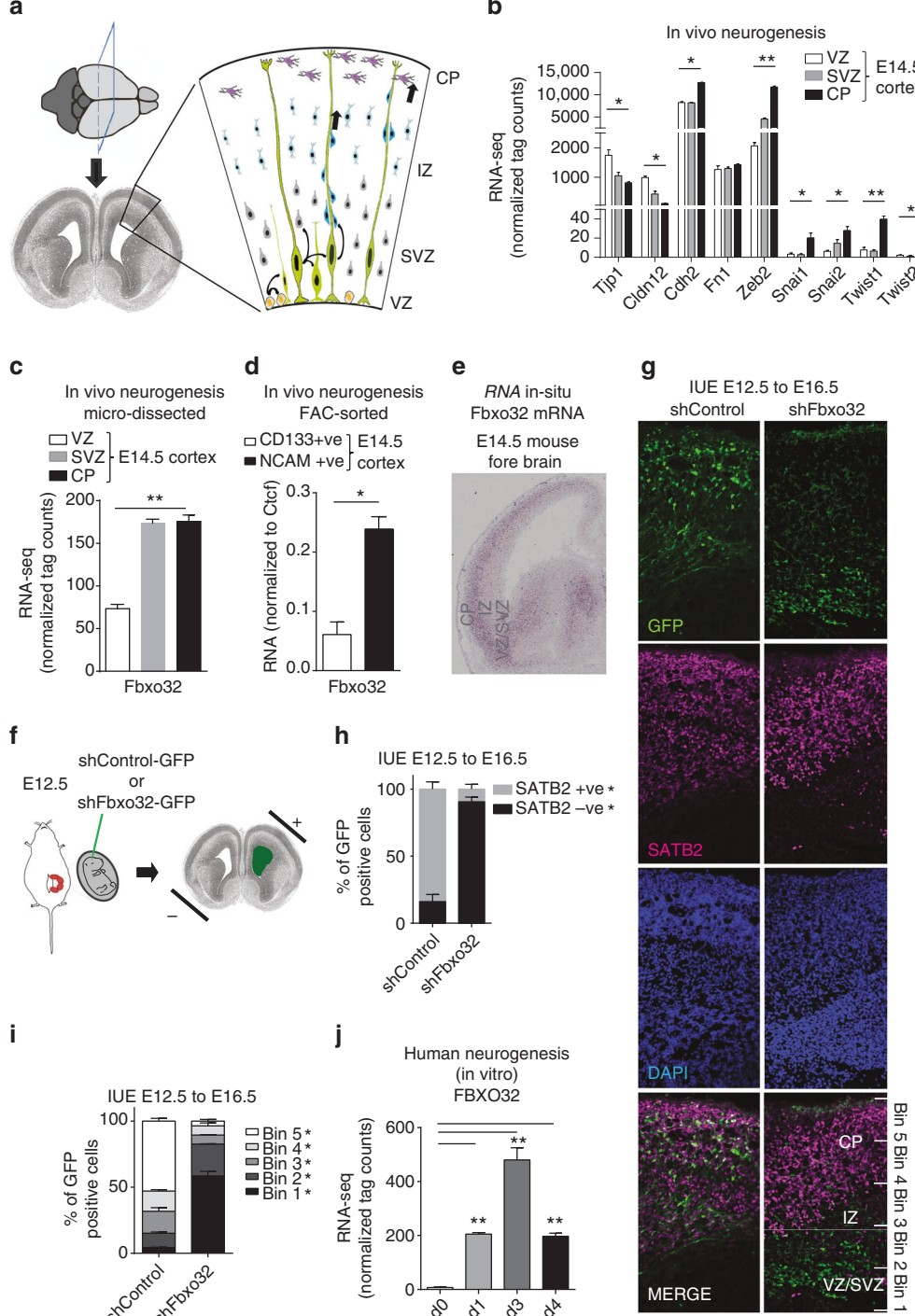

cofactors, such as CtBPs, to recruit chromatin-modifying complexes to modulate the expression of genes[13, 16, 33, 34]. In tissues, EMT is regulated via signaling by soluble growth factors (e.g., TGF-β and cytokines), composition and structure of extracellular matrix (ECM), and ECM remodeling enzymes (e.g., MMPs)[40, 48]. Recent discoveries have shed light on the importance of post-translational modifications, such as ubiquitination, during EMT[49, 50]. The K48 linked poly-ubiquitination of target proteins leads to proteasome-mediated degradation[51], whereas K63-linked ubiquitination has been implicated in the activation of several signaling pathways[52]. However, little is known about how the Ub pathway contributes to the epigenetic reprogramming that drives the gene expression program underlying EMT.

A genome-wide transcriptome analysis, in a previously characterized and established cellular model system of EMT using PMEC, identified strong transcriptional induction of F-box protein, *FBXO32*[10, 12]. FBXO32 generally acts as a substrate-recognition subunit of the SKP1–Cullin1–F-box protein E3 ligase complex for ubiquitination and was shown to have a role in muscle homeostasis[25, 53]. We found that *FBXO32* is directly modulated by TGF-β SMAD axis of EMT progression (Supplementary Fig. 2d)[32] and robustly induced during various contexts of EMT, irrespective of species and cell type. We found that FBXO32 depletion results in a phenotypic blockage of EMT in various established human and mouse cellular models, which was accompanied by a failure to gain expression of core EMT driver TFs, such as SNAIL, ZEB and TWIST, suggesting a critical requirement for FBXO32 in mediating the EMT process. Extending these findings to cancer metastasis as an example of disease EMT, we found that FBXO32 was strongly amplified in metastatic cancers, and its depletion in a NSG mouse xenograft model significantly inhibited tumor growth and metastasis. During gastrulation, pluripotent epithelial epiblast cells ingress to form the primary mesoderm through EMT[8]. Similar phenomena have been observed during later embryogenesis and organ development, such as cortical neurogenesis, where the neural stem cells from the ventricular zone undergo changes in polarity and ultimately migrate towards the cortex[54]. Our findings of an impaired neuronal EMT during brain development following FBXO32 depletion in vivo imply that FBXO32 is a critical player and member of the core EMT machinery that mediates EMT in both developmental and disease contexts. These observations are also in line with the concept that the transition of epithelial cells into mesenchymal cells, both in development and under pathological conditions, follows a conserved program[55].

FBXO32 contains a class II PDZ domain that interacts with specific sequences at the C-terminal of its target proteins, and it

has two nuclear localization signals[56]. PZD domain helps in localizing cellular elements, and regulating cellular pathways. These aspects of FBXO32 suggest that it may target transcription factors or other proteins in the nucleus for ubiquitination. Our global proteomics approach identified many FBXO32 interactors and substrates, among which CtBP1 was very strongly enriched. CtBP1 lacks nuclear localization signal and it is assumed that its hetero-dimerization with other proteins (e.g., CTBP2, KLF3) allows its translocation to the nucleus[14]. While a previous study indicated that the Pak1-mediated phosphorylation of CtBP1 causes its cytoplasmic retention[57], nothing is known about what results in its nuclear retention. Our comprehensive investigation revealed that FBXO32-dependent K63-mediated ubiquitination of CtBP1 is essential for its nuclear retention. We also find that a deletion within FBXO32 encompassing the core F-box domain results in a loss of its interaction with CtBP1. It is possible that the amino acids that were deleted in addition to the core F-box play an essential role in the interaction of FBXO32 with its substrate CtBP1. Such deletion may also affect binding of other factors directly or indirectly, which in turn could cause a loss of FBXO32 interaction with CtBP1. Future work should involve a more fine mapping of amino acid residues that are involved in FBXO32 function, including its substrate recognition. It is known that CtBP1 functions in cooperation with other chromatin remodelers such as KDM1A and transcription factors including SNAIL, ZEB[13]. Our experimental evidences revealed that FBXO32-dependent nuclear localization of CtBP1 is essential for its binding at the promoters of key EMT genes and subsequent epigenetic remodeling, including histone modifications and chromatin accessibility, thereby generating the gene expression program that drives EMT.

Among the most interesting observations following depletion of FBXO32 during EMT was the finding that the expression pattern of not only the established hallmark EMT genes but also a majority of EMT relevant genes was closely retained in an epithelial-like state. Interestingly, these data also revealed that MMPs (e.g., *MMP1*, *MMP9*, and *MMP10*), which degrade and modify the ECM and play a crucial role in EMT, require FBXO32 for their transcriptional induction during EMT. MMP1 cleaves and activates protease-activated receptor-1 (PAR-1), leading to increased migration and invasion of breast cancer cells[40]. Similarly, MMP9 mediated proteolytic activation of latent TGF-β and interleukin-8, results in a positive feed-forward loop for EMT progression[40]. An inflammatory tumor microenvironment is known to play a pivotal role in the development of cancer[58]. Notably, inflammatory cytokines such as *CXCL1/2/3/5, IL-1A/1B/ 32/6*, and many of their receptors, such as *IL1R1, CXCR1* and

---

**Fig. 5** FBXO32 is required for neuronal EMT during brain development. **a** Schematic representation of cellular morphology changes in a developing mouse cortex (E14.5) across its various layers viz. ventricular zone (VZ), sub-ventricular zone (SVZ), intermediate zone (IZ) and cortical plate (CP). **b** Normalized tag counts for key EMT markers in RNA-seq data derived from the ventricular zone (VZ), the sub-ventricular zone (SVZ), and the cortical plate (CP) of E14.5 mouse cortex. **c** The levels of Fbxo32 mRNA are shown as the average normalized tag counts derived from the RNA-seq data used in (b). **d** RT-PCR analysis of Fbxo32 in FAC-sorted cells from E14.5 mouse cortex using antibodies against CD133 and NCAM surface markers. The mRNA levels were measured relative to Ctcf via RT-PCR, and the results were plotted on the y-axis. Error bars represent the SEM of independent biological replicates (n = 3). **e** The expression pattern of Fbxo32 mRNA as visualized by in situ hybridization in E14.5 cortex. The data were extracted from the publically available data set "gene paint." **f** Graphical representation of *in utero* electroporation (IUE). **g** In utero electroporation was performed at E12.5 using plasmids containing GFP and with control shRNA or shRNA against Fbxo32, and the mice were euthanized at E16.5 for further analyses. A representative image (n = 3) of the immunofluorescence analysis performed with anti-GFP and anti-SATB2 antibodies showing the retention of GFP-positive cells below the SATB2 layer in a brain electroporated with Fbxo32 shRNA compared to a control brain. **h** A bar plot showing quantification of the migrated GFP-positive cells in control and Fbxo32-depleted mouse brains with respect to SATB2 staining. The y-axis represents the percentage of cells on or below the Satb2-stained region. The error bars represent the SEM of three independent biological replicates. *p < 0.05, Student's t test. **i** The cortex was divided into five equal bins as shown in **g**, and the number of electroporated cells was counted in each bin. The percentage of cells in each bin was plotted on the y-axis. Error bars reflect the S.E. M. of quantifications from independent brains, and significance was calculated using a t-test comparing shControl and shFbxo32 electroporated brains (* < 0.05) (n = 3). **j** The expression of FBXO32 during human neurogenesis in vitro is depicted using normalized tag counts from RNA-seq data for various stages of this process in a previous study[46]. See also Supplementary Fig. 8

*TGFBR1*, were also strongly induced by FBXO32. Altogether, these evidences emphasize the essential function of FBXO32 in establishing the microenvironment required to initiate EMT.

Further computational and experimental analysis also discovered that the genes activated by FBXO32 via CtBP1 are also targeted by ATF3. *ATF3* is induced by a variety of signals, including TGF-β, and is known to play an integral role in

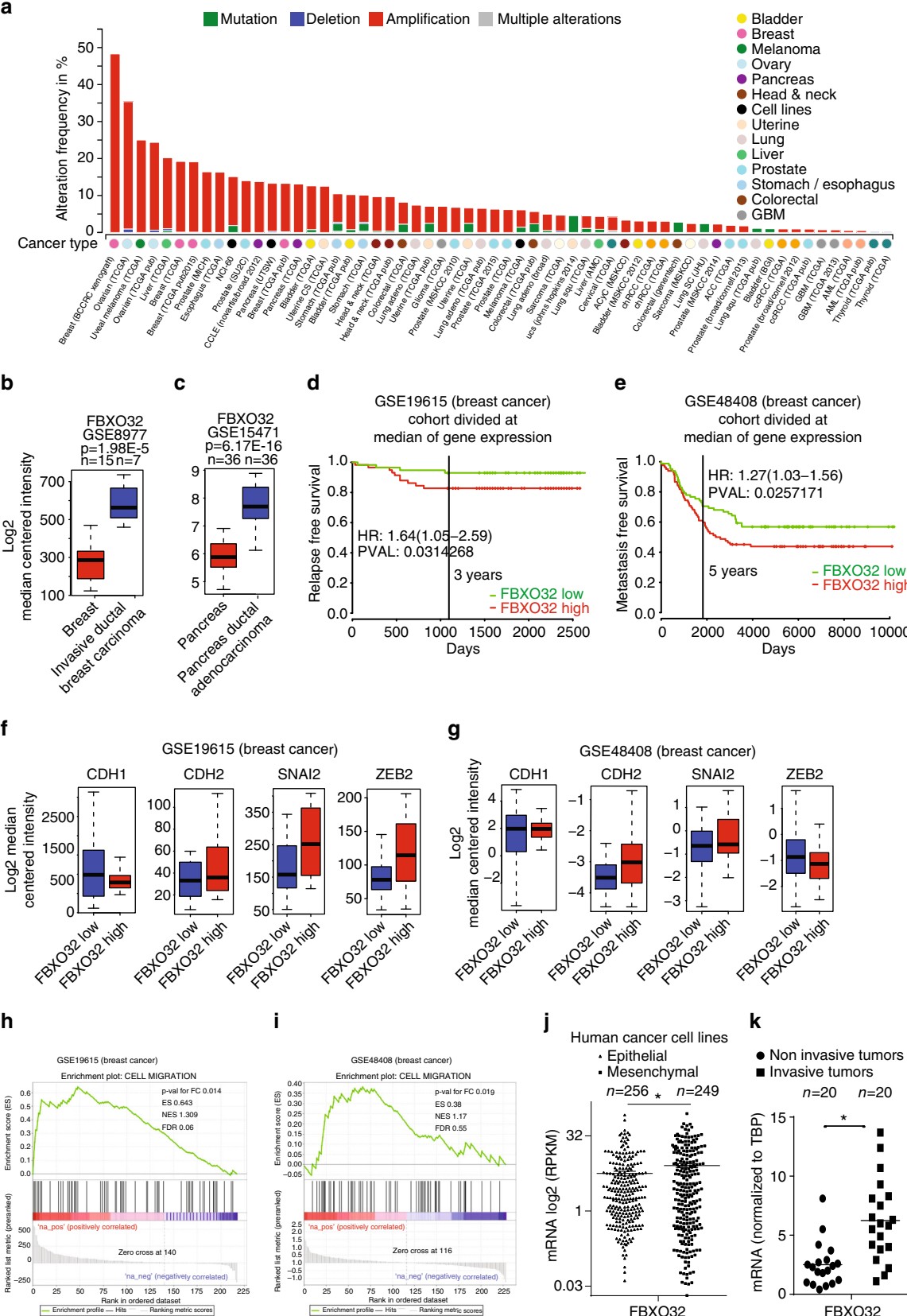

upregulating TGF-β target genes, including *SNAIL*, *SLUG*, and *TWIST*, to enhance cell motility[59]. These data suggest that other branches of signaling induced by TGF-β during EMT may converge and/or cooperate with FBXO32–CtBP1 axis to modulate expression of key EMT genes. It will be interesting to further dissect this network and unravel its relevance in fine-tuning the gene expression program underlying EMT.

We further discovered that FBXO32 not only promotes EMT but also plays a critical role in the maintenance of mesenchymal identity by regulating a distinct set of genes in each case. Since EMT plays a critical role in tumorigenicity and metastasis, we interrogated a large panel of human cancers for *FBXO32* expression and revealed a very strong induction of *FBXO32* in various tumor types compared to the matched normal tissues. Moreover, an elevated *FBXO32* expression significantly correlated with poor relapse-free, overall, and metastasis-free survival in patients with tumors of various origins, including breast, colon, lung, gastric and ovary. Moreover, *FBXO32* levels correlated positively with the levels of established mesenchymal markers and identified FBXO32-induced EMT genes, while it negatively correlated with the levels of epithelial markers in all studied tumor types. Of note, within tumors, *FBXO32* was more highly expressed in invasive tumors compared to non-invasive tumors. These findings are in full agreement with our observations of significantly reduced metastatic potential of breast cancer cells in vitro and in vivo following FBXO32 depletion. These range of findings and validations, in several model systems and primary cells from human and mouse, establish a critical role of FBXO32 in promoting tumorigenicity and metastasis in cancers.

TGF-β signaling pathways exert tumor suppressor effects in normal cells and early carcinomas. As tumors develop and progress, these protective and cytostatic effects of TGF-β are often lost due to cellular dynamics which can selectively shut down the tumor suppressive axis and TGF-β signaling then switches to promote cancer progression, invasion, and tumor metastasis[60, 61]. This could also explain why few previous studies found FBXO32 to function as a tumor suppressor in gastric and urothelial carcinoma downstream of TGF-β signaling[31, 32], while our own observations made in multiple model systems including primary EMT, metastatic cancer cell lines and clinical samples conclude that FBXO32 functions as a promoter of tumorigenicity and metastasis. Interestingly, various components of TGF-β signaling pathways are known to be primarily mutated in distinct tumor types such as colon, ovarian, and uterian carcinomas, but strongly amplified in breast, lung, pancreas, esophagus, and stomach carcinomas[61]. Our analysis showed a massive amplification of *FBXO32* in several tumor types which was very strong for most breast tumor datasets.

Altogether, our findings provide the first report on FBXO32 as a highly potent and new regulator of EMT in both development (e.g., neurogenesis) and disease (e.g., cancer metastasis) contexts of EMT. We show that FBXO32 plays a critical role in the progression and maintenance of mesenchymal identity by regulating the expression of key mesenchymal genes, including MMPs, chemokines, chemokine receptors and the extracellular matrix components. The FBXO32-mediated ubiquitination of CtBP1 promotes its nuclear localization, which is essential for its chromatin remodeling function at the target key EMT genes to mediate the gene expression program that drives EMT (Fig. 8). Together, these findings establish the function of FBXO32 as a critical regulator of EMT by governing the gene expression program underlying this process during development and disease.

## Methods

**Reagents.** The reagents used in the present study were TGF-β (rhTGF-β1 240-B, R&D Systems), DMEM (21969-035, Invitrogen), RPMI-1640 (R0883, Sigma), PBS (D8537, Sigma), trypsin (25300-054, Invitrogen), Opti-MEM (31985-047, Invitrogen), FBS (10270, Invitrogen), glutamine (25030-024, Invitrogen), MEM NEAA (100×) (11140-035, Invitrogen), Lipofectamine 2000 (11668, Invitrogen), Lipofectamine RNAiMAX (13778-150, Invitrogen), Trizol (15596026, Invitrogen), reverse transcriptase kits (K1612, Thermo Fischer), SYBR-Green PCR MasterMix (4334973, Invitrogen), Bradford reagent (5000205, BioRad), a protease inhibitor cocktail (04693132001, Roche), and a phosphatase inhibitor cocktail (04906837001, Roche).

**Inhibitors used in the study.** For chemical inhibition experiments, we used JNK inhibitors, SP600125, (10 μM) (S5567, Sigma), ERK-signaling inhibitors UO126 (25 μM) (Milipore) and p38i SCIO-469, (5 nM), (3528, r&d systems). Ubiquitination pathway inhibitor PYR-41 or 4[4-(5-Nitro-furan-2-ylmethylene)-3,5-dioxo-pyrazolidin-1-yl]-benzoic acid ethyl ester (N2915, Sigma) was added to cell culture medium to a final concentration of 10 μM, 4 h prior to collection of cells.

The cells were seeded at required densities and pretreated 30 min prior to TGF-β treatment with signaling pathway inhibitors or their solvent control DMSO at the final concentrations, mentioned above and analyzed at indicated time points. For SMAD pathway blocking, we have used on target plus smart pool siRNA (Dharmacon) targeting Smad4. This was followed by collecting the cells 24 h later for RNA analysis.

**Antibodies used for western blot, immunoprecipitation and immuno-fluorescence assays.** The antibodies used for immunofluorescence were against E-cadherin (13-1900, Invitrogen and 610182, BD Transduction Laboratories), N-cadherin (610921, BD Transduction Laboratories), ZO-1 (617300, Invitrogen), fibronectin (F-3648, Sigma-Aldrich), paxillin (610052, BD Transduction Laboratories), GFP (chicken, Aves Labs), SATB2 (Abcam, ab34735), beta-actin (C4, sc-47778, Santa Cruz), FBXO32 (sc-33782, Santa Cruz) (ab168372), ZEB1 (sc-25388 (H-102) Santa Cruz), ATF-3 (sc-188 (C-19) Santa Cruz), Cleaved caspase 3 (Asp175, 9661 S, cell signaling), FLAG tag M2 (F1804, Sigma), HA tag monoclonal antibody (c15200190, Diagenode), Myc tag (ab9106, Abcam), Ub mouse monoclonal (sc8017, Santa Cruz), Lamin B (sc-6216 (C-20), Santa Cruz), Histone H3 tri methyl K4, (ab12209, Abcam), Histone H3 tri methyl K9, (ab8898, Abcam), Histone H3 acetyl K27, (ab4729, Abcam), Alexa Fluor-488 goat anti-mouse IgG (H + L) (A11029, Invitrogen); Alexa Fluor-568 goat anti-rabbit IgG (H + L) (A11011, Invitrogen), Alexa

**Fig. 6** FBXO32 expression correlates with tumor aggressiveness. **a** Bar plot showing the frequencies of mutations, deletions, amplifications and multiple alterations in well-characterized transcriptomes and genomic datasets of different tumors of various origins available from CBioPortal. The x-axis shows various types of tumors, and the y-axis represents the percentage of tumors with a specific defect. **b**, **c** Box plots showing expression level of FBXO32 in normal vs tumor samples in two well-characterized clinical tumor datasets for breast (**b**) and pancreas (**c**) origin. Box plots represent the 25th to 75th quartiles with the bold horizontal line representing the median value. **d**, **e** A Kaplan-Meier analysis was performed for the Breast tumor datasets. Survival curves showing decreased relapse free survival (**d**) and decreased metastasis free survival (**e**), which was significantly correlated with higher FBXO32 expression. **f**, **g** Box plots showing levels of key EMT markers expression in the clinical breast tumor databases used in Fig. 5d, e, respectively, and split based on expression of FBXO32. For all such analyses, here and later, we considered the first and forth quartile of patient samples ordered based on the median expression of FBXO32. The best probe was considered for analysis, if provided by the manufacturers. Box plots represent the 25th to 75th quartiles with the bold horizontal line representing the median value. **h**, **i** Gene set enrichment analysis showing correlation between genes downregulated upon FBXO32 depletion and the genes associated to the specified GO terms, as a function of fold change between expression levels of these genes in tumors with high FBXO32 and low FBXO32 expression in the characterized tumor data sets in Fig. 5d, e, respectively. *P* value for fold change, Enrichment score (ES), Normalized Enrichment Score (NES) and FDR q-value were provided in the figure **j** The levels of FBXO32 mRNA in epithelial (n = 256) and mesenchymal (n = 249) cancer cell lines are shown as the FPKM derived from the publically available RNA-seq data (C Klijn et al, 2015) and the results were plotted on the y-axis. **k** The levels of FBXO32 mRNA in non-invasive (DCIS) (n = 20) and invasive tumor samples (n = 20) were measured relative to TBP via RT-PCR, and the results were plotted on the y-axis. *p < 0.05, Student's t test. See also Supplementary Figs. 9 and 10

Fluor-633 goat anti-rat IgG (H + L) (A21094, Invitrogen), and Alexa Fluor-633 phalloidin (A22284, Invitrogen), which was used to stain F-actin.

**Cell culture.** HMECs were obtained from Lonza and cultured according to the manufacturer's guidelines. Other cell lines were obtained from ATCC and cultured in the following media: MDA-MB-231: DMEM, 10% FBS; MDA-MB-361: DMEM, 20% FBS; BT549: RPMI-1640, 10% FBS, 0.001 mg/ml bovine insulin; MCF7: DMEM, 10% FBS, 0.01 mg/ml bovine insulin; HEK293T: the same medium as NMuMG. HMLE cells were a kind gift from Christina Scheel (Helmholtz Zentrum Munich) and cultured in LONZA primary cell culture medium. A subclone of NMuMG cells was grown in DMEM supplemented with 10% FBS, 2 mM L-glutamine and 1× non-essential amino acids[62]. All the cells were cultured at 37 °C with 7% $CO_2$ in a humid incubator. For TGF-β time-course experiments, NMuMG, HMEC and HMLE cells were treated with 2, 5 and 5 ng/ml TGF-β, respectively, for the indicated times. TGF-β was replenished and the medium was changed every 2 days.

**siRNA-mediated knockdown.** For all siRNA-mediated knockdown experiments, the cells were seeded at the same starting density and transfected every second day with ON-TARGET plus single siRNA or SMART pool siRNAs (i.e., a mixture of 4 siRNAs provided as a single reagent) (Dharmacon). For siRNA transfections, Lipofectamine RNAiMAX (Invitrogen, 13778-150) was used according to the manufacturer's instructions. For experiments during TGF-β-induced EMT, TGF-β induction was performed at the same time that siRNA was added to avoid indirect effects due to loss of protein function. The sequences of the siRNAs are provided in Supplementary Data 1.

**Mammosphere culture.** For primary sphere formation, single cells were plated in ultralow attachment plates (Corning) at a density of 20,000 viable cells/mL in serum-free mammary epithelial growth medium composed of DMEM supplemented with 1% L-glutamine, 1% penicillin/streptomycin, 30% F12, 2% B27 (Invitrogen), 20 ng/mL EGF (peprotech, AF-100-15) and 20 ng/mL bFGF

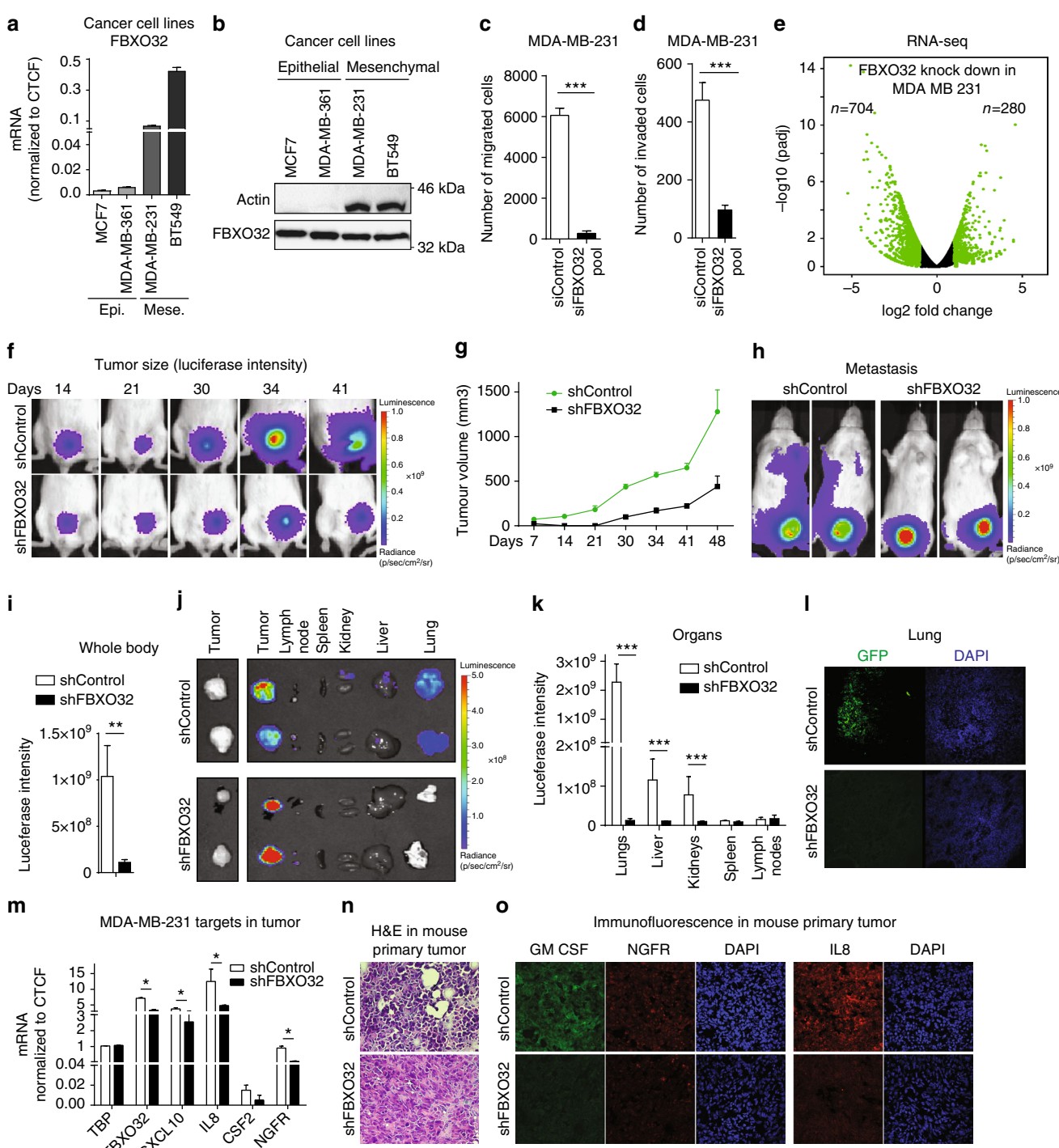

(peprotech, 100-18B). The medium was made semi-solid by the addition of 0.5% Methylcellulose (Sigma) to prevent cell aggregation. Mammospheres were collected by gentle centrifugation (200 g) after 10 days and dissociated enzymatically (10 min in 0.05% trypsin, 0.53 mM EDTA, Invitrogen) and mechanically, using a fire-polished Pasteur pipette. The cells obtained from dissociation were sieved through a 40-µm sieve and analyzed microscopically for single-cellularity. For the secondary sphere formation, 1000 cells/ml were plated and colonies were monitored by a microscope at a regular interval.

**Immunofluorescence assay**. The cells were grown on coverslips, fixed with 4% paraformaldehyde in PBS and permeabilized with 0.2% Triton X-100 for 15 min at room temperature. Subsequently, the cells were blocked with 10% goat serum, 5% FCS and 0.5% BSA in PBS for 20 min and were incubated with primary antibodies at 4 °C overnight. The cells were then incubated with fluorochrome-labeled secondary antibody or Phalloidin-633 for 1 h at room temperature. The coverslips were counterstained with Hoechst, mounted with immu-mount and imaged using a confocal laser-scanning microscope. The data were processed by ImageJ software.

**Immunohistochemistry for analysis of mouse cortex**. The isolated E16.5 embryonic brains were immediately fixed for 6 h in 4% PFA in PBS at 4 °C. The brains were then cryoprotected in 10% sucrose for 2 h and then in 30% sucrose (in PBS) overnight, embedded in Tissue-Tek, stored at −20 °C and cryo-sectioned at 12 µm. Sections on coverslips were preblocked with 2% BSA and 0.5% Triton (in PBS) for 1 h. Primary antibodies (Satb2, 1:500, and anti-GFP (chicken, Aves Labs, 1:1000)) were applied in blocking solution overnight at 4 °C. Fluorescent secondary antibodies were applied according to the manufacturer's protocol (Life Technologies). The coverslips were counterstained with Hoechst, mounted with immu-mount and imaged using a confocal laser-scanning microscope (Leica SP5). The data were processed with ImageJ software.

**Quantitative real time PCR**. mRNA levels were quantified as previously described[12]. In brief, total RNA was prepared using Trizol (Invitrogen) or a SurePrep TrueTotal RNA Purification Kit (Fisher Scientific) and was reverse transcribed with a First Strand cDNA Synthesis Kit (Fermentas). The transcripts were quantified via PCR using SYBR green PCR MasterMix (ABI) on a ViiA7 PCR machine (Life Technologies). Human or mouse CTCF/Ctcf and TBP/Tbp primers were used for normalization. The sequences of all of the primers used in this study are listed in Supplementary Data 1.

**Immunoblotting**. The cells were lysed in RIPA buffer, and protein concentrations were quantified using Bradford reagent. Equal amounts of protein (30 µg) were boiled in 6 × SDS-PAGE loading buffer, subjected to polyacrylamide gel electrophoresis, transferred to a PVDF membrane and probed with the appropriate antibodies. All uncropped western blot images can be found in Supplementary Fig. 12.

**Chromatin Immunoprecipitation assay**. The cells were cross-linked in medium containing 1% formaldehyde for 10 min at room temperature, neutralized with 0.125 M glycine, scraped, and rinsed twice with PBS. The pellets were suspended in buffer L1 (50 mM Hepes KOH, pH 7.5, 140 mM NaCl, 1 mM EDTA pH 8.0, 10% glycerol, 5% NP-40, and 0.25% Triton-X 100) and incubated for 10 min at 4 °C. The cells were then suspended in buffer L2 (200 mM NaCl, 1 mM EDTA pH 8.0, 0.5 mM EGTA pH 8.0, 10 mM Tris pH 8.0) for 10 min at room temperature. Finally, the pellet was suspended in buffer L3 (1 mM EDTA pH 8.0, 0.5 mM EGTA pH 8.0, 10 mM Tris pH 8.0, 100 mM NaCl, 0.1% Na-deoxycholate, 0.17 mM N-lauroyl sarcosine) containing protease inhibitors and was incubated at 4 °C for 3 h following sonication using Bioruptor plus (Diagenode). Sixty micrograms of chromatin was incubated overnight at 4 °C with 2 µg of the antibodies targeting

H3K4me3, H3K9me3, H3K27ac (Abcam) or Myc (Santa Cruz) and then incubated with preblocked beads for 4 h. Finally, the beads were washed twice with L3 and once with 1 ml of DOC buffer (10 mM Tris (pH 8.0), 0.25 M LiCl, 0.5% NP-40, 0.5% deoxycholate, 1 mM EDTA), and the bound chromatin was eluted in 1% SDS/ 0.1 M NaHCO3. This followed treatment with RNase A (0.2 mg/ml) for 30 min at 37 °C and then with proteinase K (50 µg/ml) for 2.5 h at 55 °C. The crosslinking was reversed at 65 °C overnight with gentle shaking. The DNA was purified by phenol-chloroform extraction followed by ethanol precipitation and was recovered in TE buffer.

**Mass spectrometry analysis**. Peptide fractions were analyzed using a quadrupole Orbitrap mass spectrometer (Q Exactive Plus, Thermo Scientific) equipped with a UHPLC system (EASY-nLC 1000, Thermo Scientific), as described[63]. Peptide samples were loaded onto C18 reversed-phase columns and eluted for 2 h with a linear gradient of acetonitrile from 8 to 40% containing 0.1% formic acid. The mass spectrometer was operated in data-dependent mode with automatic switching between MS and MS2 acquisition. Survey full scan MS spectra (m/z 300–1650) were acquired in the Orbitrap. The 10 most intense ions were sequentially isolated and fragmented by higher-energy C-trap dissociation (HCD)[64]. Peptides with unassigned charge states or charge states less than +2 were excluded from fragmentation. The fragment spectra were acquired in the Orbitrap mass analyzer.

Peptide identification: the raw data files were analyzed using MaxQuant (development version 1.5.2.8)[65]. Parent ion and MS2 spectra were searched against a database containing 88,473 human protein sequences obtained from the UniProtKB released in December 2013 using the Andromeda search engine[66]. The spectra were searched with a mass tolerance of 6 ppm in MS mode, 20 ppm in HCD MS2 mode, strict trypsin specificity and allowing up to 3 miscleavages. Cysteine carbamidomethylation was searched as a fixed modification, whereas protein N-terminal acetylation and methionine oxidation were searched as variable modifications. The data set was filtered based on posterior error probability (PEP) to arrive at a false discovery rate below 1%, estimated using a target-decoy approach[67].

**In vivo tumorigenicity assays and imaging**. NOD *scid* gamma (NSG) mice were bred and maintained under SPF conditions in the Translational Animal Research Center at the University Medical Center Mainz. We did preliminary experiments to determine the need for mice sample size and duration of experiment. The MDA-MB-231 cells with stable expression of GFP and a luciferase reporter were used for generating stable cell lines with either control shRNA or shRNA against FBXO32. Briefly, cells were counted and resuspended in a 1:1 (v/v) mixture of PBS and Matrigel (BD Biosciences). Ten-week-old female mice were injected unilaterally with $2 \times 10^6$ cells in 100 µl of 50:50 Matrigel/PBS into the fourth abdominal fat pad via subcutaneous injection at the base of the nipple. Tumor growth was monitored externally using Vernier calipers for up to 48 days. The maximum size of the tumor that was permitted in the mice was 2700 mm³ according to ethical approval policy of Translational Animal Research Center at the University Medical Center Mainz. The tumor volume was calculated as follows: tumor volume (mm³) = length × (width)² × 0.5. Necropsies were performed to identify macro-metastases. Primary tumors and organs were immediately frozen in liquid nitrogen and stored at −80 °C until use.

In vivo bioluminescence imaging of tumor-bearing mice and their organs was performed at day 50 using an IVIS Lumina imaging system (Perkin Elmer). Briefly, mice were anesthetized with isoflurane and an aqueous solution of D-luciferin-K⁺ salt (150 mg/kg body weight) (Perkin Elmer) was injected intraperitoneally. Five minutes after the injection, the mouse was placed onto the imaging chamber of IVIS, and photons acquired with an integration time of 10 s were presented as color-scaled images using IVIS Living Image Software (version 4.3.1) (Perkin Elmer). For organ imaging, mice were euthanized after luciferin injection, and the dissected organs were imaged as described above.

---

**Fig. 7** FBXO32 is crucial for the maintenance of mesenchymal identity and promotes tumorigenicity and metastasis in vivo. **a** The levels of FBXO32 mRNA in two human epithelial cell lines (MCF7 and MDA-MB-361) and two mesenchymal cell lines (MDA-MB-231 and BT549) were measured relative to CTCF using RT-qPCR. **b** Western blot analysis of FBXO32 in cell lines described in **a**. β-ACTIN acted as a loading control. **c, d** Migration **c** and invasion (**d**) assays in MDA-MB-231 cells transfected with either control siRNA or siRNA against FBXO32 for 4 days. **e** Volcano plots showing significantly differentially expressed genes upon FBXO32 depletion in the MDA-MB-231 cell line. **f, g** Representative luciferase images of tumor-bearing mice at various time points in the mice (**f**) and line graph (**g**) showing quantification of tumor size growth in mice (n = 6 per condition) analyzed at various time points after mammary fat pad injection of MDA-MB-231 cells stably expressing either control shRNA or shRNA against FBXO32 alongwith GFP and a luciferase reporter. The x-axis represents days post-injection, and the y-axis represents the size of tumors in mm³. **h, i** Representative whole-body luciferase scanning (**h**) along with quantification (**i**) showing primary and metastatic tumors (from **f** to **g**) (n = 6 per condition). The primary tumor in the control mouse gets bigger and become necrotic leading to a decrease in the fluorescence intensity at the site of transplant. **j, k** Similar analysis as in **h, i** showing luciferase scanning image (**j**), along with quantifications (**k**) of various organs from those mice to detect metastatic tumors. **l** Lung sections were collected from the mice in the above experiment after euthanasia, and immunofluorescence images were obtained using an anti-GFP antibody to detect migrated cells. **m** mRNA levels of key genes deregulated upon FBXO32 depletion was validated in primary tumors obtained from the mouse used in Fig. 6i were measured relative to CTCF. **n, o** H&E (**n**) and immunofluorescence images (**o**) for key cytokines were measured in primary tumor obtained from the mouse used in **h**. Scale bar, 100 µm. Error bars represent the SEM of three independent biological replicates (n = 3). ***p < 0.001, Student's t test. See also Supplementary Fig. 11

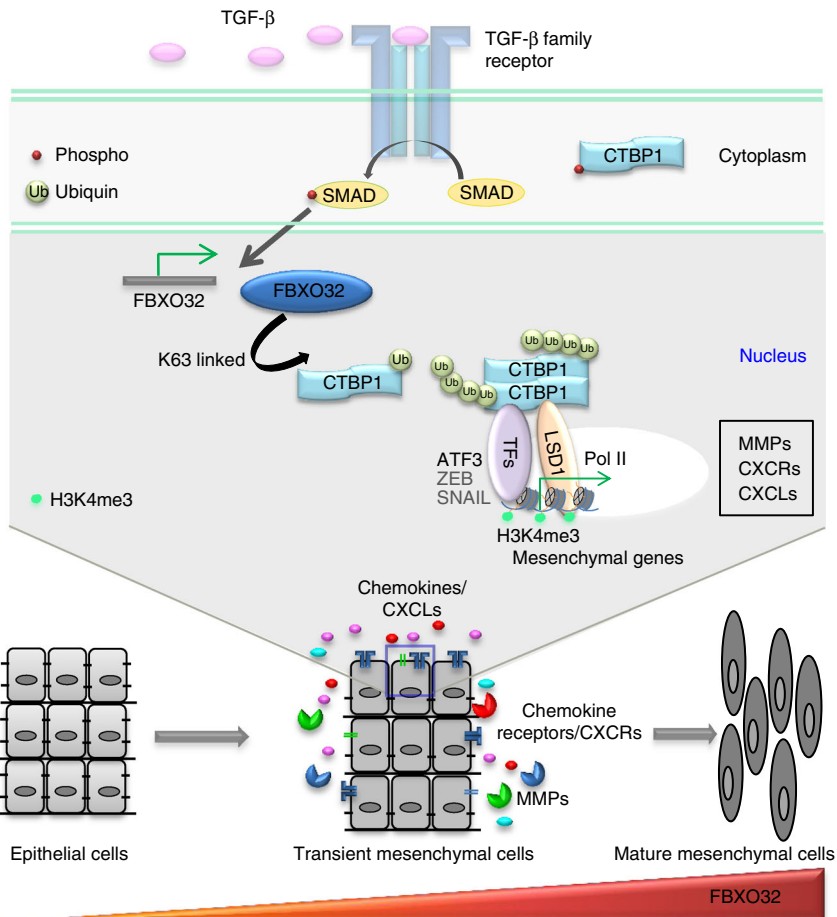

**Fig. 8** FBXO32 acts as a master regulator of EMT by governing the gene expression program underlying this process. The model showing roles of FBXO32 during EMT. The onset of EMT crucially relies on SMAD-mediated transcriptional changes, and FBXO32 is one of the ubiquitin ligases upregulated during this process. FBXO32-mediated-K63 ubiquitination of CtBP1 leads to the nuclear retention of CtBP1 and so it can be recruited by various transcription factors like ATF3 to assist in chromatin remodeling. This leads to alterations in the transcription of key EMT and migration promoting genes, such as Matrix metalloproteinases (MMPs), chemokines and chemokine receptors. Such function of FBXO32 is essential for EMT in both development (e.g., neurogenesis) and disease (e.g., cancer metastasis) contexts, establishing this as a component of the EMT core machinery. Moreover, FBXO32-mediated induction of cytokine secretion acts as a feed forward loop for the EMT and further enhances the epithelial to mesenchymal transition process

Animal maintenance and experiments were performed under an approved protocol in accordance with the animal care guidelines of Johannes Gutenberg University.

**In utero electroporation**. In utero electroporation experiments were performed essentially as previously described[45]. All the experimental procedures were conducted in accordance with European, National, and Institutional Guidelines for animal care. Timed-pregnant (Theiler stage 20 = E12.5) C57BL/6 mice were anesthetized with isoflurane (2.5% via mask, Forene®, Abbod), and carprofen (4 mg/kg body weight, Rimadyl, Pfizer) was administered subcutaneously as an analgesic. After the abdominal cavity was opened, the embryos were carefully exposed, and 1 µl of colored solution containing 1 µg of the p.SUPER-GFP shRNA plasmid expressing shRNA against Fbxo32 or control shRNA was injected into one of the lateral ventricles. Using specialized platinum electrodes (Nepagene CUY 650 P), the appropriate voltage was applied (50 ms, interval 950 ms, 5 pulses). After electroporation, the uterine horns were returned to the abdominal cavity.

**Cell survival assay**. The cells were pre-depleted with siRNA (a pool of 4 siRNAs targeting the same gene, SMARTpool, Dharmacon) for two days, and 40,000 cells per 6 cm were plated and cultured for 6 days. Fresh medium along with siRNA was added every other day. The cells were counted, or bright-field images were obtained, or the cells were fixed with 4% PFA. To visualize the fixed colonies, the cells were stained for 10 min with a solution containing crystal violet and 10% ethanol, and the plates were washed in water until the excess dye was washed away. Photographs of the plates were obtained for later analysis.

**Wounding migration assay**. The cells depleted of FBXO32 were seeded at equal densities, and a scratch wound was generated using a 10-µl pipette tip on confluent

cell monolayers grown in six-well culture plates. The cells were then washed with fresh medium to remove floating cells. Bright-field images were obtained at 20x magnification after 18 h of wounding.

**Migration and invasion assays**. Migration assays were performed as previously described[12]. Briefly, 10,000 cells were seeded in 2% FBS/DMEM (Sigma) into the upper chamber of a 24-well Transwell migration insert (pore size 8 µm; Falcon BD). The lower chamber was filled with 20% FBS/DMEM. After 16 h of incubation at 37 °C, the cells in the upper chamber were removed using a cotton swab, and the cells that had traversed the membrane were fixed in 4% paraformaldehyde/PBS and quantified by DAPI staining using a fluorescence microscope and ImageJ software. For the invasion assays, transwells were coated with 0.5 mg/ml Matrigel in serum-free medium overnight, and $1.5 \times 10^5$ cells were used for the assays.

**IP and ubiquitylation assays**. HEK-293T cells were co-transfected in various combinations with expression vectors for HA-tagged Ub, Myc-tagged CtBP1 and Flag-tagged FBXO32 using Lipofectamine® 2000 Transfection Reagent (Thermo scientific). For inhibition of ubiquitination pathway, UBEI/PYR-41 was added at a final concentration of 10 µM for 4 h. All experimental cells for ubiquitination assay were treated with 10 µM of MG132 for 4 h prior to lysis. Cells were lysed in lysis buffer (50 mM HEPES pH 7.5, 150 mM NaCl, 5 mM EGTA, 1.5 mM MgCl2, 1% Glycerol, 1% Triton X-100) and centrifuged for 1 h to collect the soluble fraction. Tagged proteins were immunoprecipitated for overnight at 4 °C with the appropriate antibody (5 µg/IP). Pre-blocked beads were added to the lysate and incubated for 3 h at 4 °C. The bound beads were washed two times with lysis buffer and two times with wash buffer (50 mM Tris-HCl pH 7.5, 150 mM NaCl, 5 mM EDTA, 0.1% Triton X-100) and analyzed by immunoblot analysis. For all IP experiments, immunoprecipitated material was eluted with an equal volume of elution buffer

and same amount of samples were loaded on gel for immunoblot analysis. To assess ubiquitylation, 293T cells were transfected with various tagged expression vectors in different combinations (as outlined in the manuscript figures) in the presence of TGF-β for 4 days. Proteins lysate were precipitated with anti-Myc antibody to pull down CtBP1 and analyzed by immunoblot using anti-HA antibody to detect Ub.

**RNA sequencing data analysis**. RNA-seq data were generated using Illumina sequencing. The reads were aligned to the mouse genome (mm9) using TopHat[68] (version 2.0.9) with the default options. After library size normalization using DESeq[69], expression was quantified and expressed in reads/kilobase of transcript per million mapped reads (RPKM) using cufflink (version 2.1.1)[68]. A differential expression analysis was performed using the DESeq package with an FDR cutoff of 0.1[69]. Gene set enrichment analysis was performed using GSEA Preranked module[70].

**Data availability**. The authors declare that all the data supporting the findings of this study are available within the article and its Supplementary Figure files or from the corresponding author upon reasonable request. The RNA-seq data generated during this study have been deposited in the Genbank nucleaotide database (GEO) under accession code GSE77408. The proteomic mass spectroscopy data have been submitted as an excel file (Supplementary Data 2). In addition to these datasets, we had also used a number of publically available data sets (GEO accession codes: GSE54133, GSE62146, GSE8977, GSE15471, GSE19615, GSE48408, GSE19615, GSE48408, GSE30611, GSE95983, GSE32863, GSE31210, GSE6764, GSE62254, GSE31210, GSE9891, GSE28814, GSE28722).

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

## Acknowledgements

We thank the members of the Tiwari lab for their cooperation and critical feedback throughout the course of this study. The support from the Core Facilities of the Institute of Molecular Biology (IMB), Mainz, is gratefully acknowledged, especially the microscopy, cytometry, genomics and bioinformatics core facilities. We especially thank Prof. Robert Nitsch and Nikolai Schmarowski from Institute for Microscopic Anatomy and Neurobiology University Medical Center, Johannes Gutenberg University in Mainz for help with the in utero electroporation experiments and Prof. Marcus Schmidt from Department of Obstetrics and Gynecology, Johannes Gutenberg University in Mainz for providing clinical samples. We are also grateful to Dr. Christina Scheel (Helmholtz Zentrum München) for providing HMLE cells. This study was exclusively supported by the Wilhelm Sander Stiftung 2012.009.1 and 2012.009.2 to V.K.T. The research conducted in the laboratory of P.B. is supported by the Emmy Noether grant BE 5342/1-1 and the Marie Curie grant CIG 630763.

## Author contributions

S.K.S.: Designed and performed the experiments, analyzed the data and wrote the manuscript. N.T.: Initiated and supervised the study, performed the experiments, analyzed the data and wrote the manuscript. A.P.: Performed computational analysis. Y.Z.: Performed the experiments. M.B. and P.B.: Assisted with mass-spectrometry experiments and analysis. S.S. and M.D.: Provided help with in vivo tumorigenicity assays and imaging. V.K.T.: Designed and supervised the study, analyzed the data and wrote the manuscript. All authors read and approved the final manuscript.

## Additional information

**Competing interests:** The authors declare no competing financial interests.

