## [Peer Review File · Nature Communications]

Reviewers' Comments:

Reviewer #1 (Remarks to the Author)

In the current manuscript of Sahu et al. the authors screen for novel mediators of EMT and identified FXBO32, which is required for TGFbeta mediated induction of EMT. The analysis shows that FXBO32 primarily interacts and ubiquitinates CtBP1 and that ubiquitination of CtBP1 alters its function and localization. CtBP1 ubiquitination is required to induce target gene expression in cooperation with ATF3. Analysis of human cancer specimen revealed that FXBO32 expression is increased in tumors and that high levels correlates with poor prognosis. Moreover, knockdown of FXBO32 in aggressive breast cancer cells decelerated tumor growth upon orthotopic transplantation and reduced metastatic burden. The authors conclude that FXBO32 is essential for EMT during development and disease.

The role of the EMT process in tumorigenesis and metastasis was recently challenged (Zheng et al., Nature 2015; Fischer et al., 2015), indicating that our understanding about the underlying mechanism of EMT and malignancy is still limited and views are controversial. Moreover, the function of posttranscriptional modifications in EMT was greatly underestimated and even less analyzed. Therefore, Sahu and colleagues are showing a novel and important mechanism about how EMT inducing signals can be processed and executed within the cell. The experiments are well designed, performed and analyzed and the manuscript is written in a concise way. However, I raised some important point that need to be addressed before publication.

The authors describe an EMT cascade that is initiated by FXBO32 upregulation, resulting in CtBP1 ubiquitination, that induces CtBP1 interaction with ATF3 complex formation for transactivation of target genes. Although most experiments support this hypothesis some conclusions are not convincing and need additional experimental evidence. Moreover, the in vivo analysis of FXBO32 in tumorigenesis leaves some important questions unanswered and can easily be improved by a more thorough analysis.

Specific points:

1. A general conceptual problem is that the authors claim that they observe a reversion of the EMT phenotype when for example FXBO32 is knocked down. However, siRNA transfection is performed simultaneously with TGFbeta treatment. So very likely, the cells are blocked in an epithelial state and do not revert to it (E.g. on page 7, lines 137, 138, and following pages). This holds true for the differentially regulated genes on page 8: The term "up- and downregulated genes" is not precise enough. In fact, these genes are likely prevented from being regulated by TGFbeta and cells remain in an epithelial state. For the same reason the expression "mesenchymal cells depleted of FXBO32" on page 8, line 170 is also misleading, as these cells do not become mesenchymal.
2. Figs. S2g, Fig. 6b: With the exception of these two figures, the function of FXBO32 is mainly concluded from its differential regulation on mRNA level. It would be very important to demonstrate endogenous expression on protein level by WB or IF as well, especially its upregulation during EMT and reduction during siRNA knockdown (NMuMG, MDA-MB-231). Refs 31, 32 used antibodies that worked in WB and IHC, e.g. ab74023.
3. The analysis on K63/K48 ubiquitination that results in the conclusion that FXBO32 induces CtBP1 ubiquitination via K63 is not convincing! This needs to be supported by additional experiments, e.g. by including other known substrates of FXBO32, inhibition of ubiquitination (inhibitor UBE1-41/PYR-41) or using a non-ubiquitinatable version of CtBP1. Alternatively, the statements in the results, discussion and especially in the summary of the introduction sections needs to be weakened. In Fig. 3c, g it is not clear which of the bands represents CtBP1. Does it overlap with the size of the band in the myc-CtBP1 blot? The authors should include positions of molecular weight standards to all blots.
4. page 34, Fig. 4l: The identification of ATF3 as candidate to cooperates with CtBP1 for

transcriptional activation is not convincing. In particular, for ChIP at least day 0 untreated control should be included in the TGFbeta experiment. In addition, siFXBO32 or siCtBP1 need to be done in this experiment. In general ChIP experiments should be consistently displayed as percentage of input and an IgG control sample included for some experiments, especially in Fig. 4I. In line with the conclusion about ATF3: Fig. S6b shows ATF3 binding at the FXBO32 in ChIP-Seq data sets. However, an ATF3 knockdown does not affect FXBO32 expression as shown in Fig. 4k. How is that explained? Presumably, ATF3-mediated gene regulation is either more complex or it is the wrong candidate. Is gene regulation by known CtBP1 protein complexes (with Snail, Zeb1/2) changing upon FXBO32 knockdown? Are E-boxes found in the analyzed promoter regions in downregulated gene promoters? Is it possible that FBXO32 is turning EMT-TFs Zeb1/2, Snail, Slug into activators?

5. The authors should try to confirm the FBXO32/CTBP1 interaction by IP of endogenous proteins in NMuMG, HMEC or MDA cells, maybe in response to TGFbeta, when this interaction is induced. This would greatly improve the conclusion from the artificial overexpression experiments in 293 cells (do 293 cells respond to TGFbeta?).

6. Fig. 3d: It is puzzling that CtBP1 levels are decreasing during TGFbeta treatment, as the authors claim that FBXO32 is upregulated upon TGFbeta mediated EMT (as observed in HMECs) and overexpression of FBXO32 is increasing CTBP1 levels. This should be discussed and can be supported by showing an FBXO32 WB!

7. Fig. 6: As mentioned above, the in vivo analysis of FXBO32 in tumorigenesis could be more thorough: Valuable information can be extracted from sections of primary xenograft and metastatic lungs stained by H&E, for EMT markers and for some candidates like MMPs, HAS2, etc. qRT-PCR of same (as in HMECs) or new top candidates in MDA cells upon FBXO32 knockdown (including EMT markers) should be done. Are the same genes affected in MDA-MB-231 FXBO32 knockdown cells as in the initial screen? Is CtBP localization changing in MDA-MB-231 shFXBO32 cells?

8. page 15, Fig 6c: Colony formation was analyzed here, but to show what? In the methods section a cell survival assay is described which is not a clonogenic colony forming assay. The way it was performed, it simply shows cell viability upon trypsinization. I suggest to include a more thorough analysis of proliferation and cell death, especially since it is known that FXBO32 is regulating apoptosis as well and increased apoptosis might challenge the xenograft experiment. The results of this will also affect Figs 6d,e. Maybe cells do not survive during plating for migration/invasion and therefore fewer cells are found on the other side of the transwell filter for analysis in the FXBO32 situation? Moreover, a real sphere forming assay in presence of methylcellulose and absence of serum would be very valuable as an in vitro tumorigenicity assay.

9. The gene set enrichment analysis in Fig. 5h,I, Fig. S7 is lacking NES, FDR and p-values to make the analysis valuable. Are these changes significant? How were the GSEAs generated? By multiple analysis of many genesets simultaneously (how many and which ones?) or by individual comparison, as this will affect the outcome of the analyses?

10. The results of Fig. 5b-e, S7, S8 are somewhat controversial with e.g. Refs 31 and 32. In these publications authors describe that FXBO32 acts more like a tumor suppressor in gastric cardia adenocarcinoma and urothelial carcinoma. These discrepancies should be discussed, maybe in line with the dual role of TGFbeta signaling during tumor progression (early: inducer of differentiation -> tumor suppressor; late: inducer of EMT and stemness -> tumor promoter) and frequent loss of SMAD4 in many tumors.

Minor points:

1. Fig S9b: MDA cells are already very mesenchymal, why are they treated with TGFbeta? One would expect that FBXO3 knockdown would induce MET in steady state conditions without TGFbeta. Measuring differentially regulated genes in untreated MDA shFXBO32 cells would be helpful to dissect this.

2. Did the authors check whether Fbxo32 is also regulated during other TGFbeta expression profiles (e.g. by the Christofori lab?)

3. page 40, model Fig. 7: The epithelium looks weird, like it is stratified with multiple basement membranes

4. Fig. S2d: inhibitors are not specified in the manuscript methods
5. IP and ubiquitination assays were not described in methods
6. Fig. 6i: please include the day of analysis for metastasis
7. Fig. S5b is redundant and should replace Fig 4b
8. page 16, line 337: Figs. S9d,e are not present in the current version
9. p. 9, line 292: "FBXO32 levels correlated positively with the levels of established mesenchymal markers and negatively with the levels of epithelial markers in different types of tumor examined (Fig S7e-f)." Only CDH1 was shown as epithelial marker. Either others should be included or sentence rephrased.
10. page 18, line 372: "nude mouse xenograft". In the methods and results it is stated that NSG mice were used.
11. Typos: line 310: "NSG mice model" -> NSG mouse model; line 461: "A previously described, a subclone of..." -> as previously described; legend to Fig. 1: "(e-f) ...the levels of BXO32 and key" -> FBXO32; p. 34, Fig 4j: containingig.

Reviewer #2 (Remarks to the Author)

In the manuscript (NCOMMS-16-29334) entitled "FBXO32 mediates transcriptional program critical to promote microenvironment underlying epithelial-mesenchymal transition", the authors discovered that FBXO32, an E3 ubiquitin ligase, acts as an upstream regulator of epithelial-mesenchymal transition (EMT) by governing the gene expression, and might play important roles in cancer metastasis and brain development. This study provided an in-depth analysis of FBXO32 on its function and underlying mechanisms, which might contribute to the understanding of EMT during disease and development. The observation is interesting, but the authors did not provide solid evidences for the novel mechanism of FBXO32 in EMT, particularly for the functional relationship between FBXO32 and CTBP1. The study is premature, and the data cannot fully support the conclusions. The major concerns are as following:

Major comments:

1. The decrease of some of EMT marker genes, such as N-cadherin, Phalloidin and Paxillin, after FBXO32 depletion (shown in Figures 1D and s2i) is not that severe as authors claimed in the text, and these EMT markers are better organized in categories like epithelial or mesenchymal for clarity. How about their mRNA/protein levels?
2. While most of the experiments were done by using siRNA knockdown, a rescue experiment is required to confirm the function of FBXO32 in EMT, i.e., use siRNA-resistant FBXO32.
3. The author performed IP-MS to identify that CTBP1 as a partner protein with FBXO32, but the experimental design has not been described in detail. This experiment is better showed in TGF- β treated cells with endogenous FBXO32 pull-down. Additionally, all the Co-IP experiments were carried out using overexpressed plasmids, which cannot account for the endogenous interaction between FBXO32 and CTBP1 proteins during EMT process. A native Co-IP experiment should be provided.
4. The author found that FBXO32 affected the CTBP1 nuclear retention via K63-linked ubiquitination, instead of K48. Whether this K63-dependent ubiquitination is essential for FBXO32-mediated EMT and CTBP1 nuclear retention is not known. Thus, K63 site-mutation should be performed to confirm the direct functional relationship between K63-dependent ubiquitination and CTBP1 nuclear retention.
5. In Figure 4, the authors showed that CTBP1 depletion shared many EMT related genes with FBXO32 knockdown. FBXO32 depletion results in a lower binding of CTBP1 on EMT gene promoters. The authors concluded that FBXO32 mediates transcriptional activation of key EMT genes via nuclear CTBP1 targeting to their promoters and epigenetic remodeling, but these data cannot fully support the conclusion. As we mentioned above that the functional role of CTBP1 nuclear retention in EMT has not been fully characterized, the author need to overexpress FBXO32 in CTBP1 depleted cells to examine how the EMT is affected (like in Figure 4c).
6. Did FBXO32 promote tumorigenicity and cancer metastasis (Figures 5 and 6) through CTBP1?

Minor points:

1. The immunohistochemical pictures showing cells in Figures 1D and s2i are not convincing enough. Authors should represent Z-stack and/or 3D views.
2. Please specify the bands for Control in Figures 2c and 2d.
3. The background of Western-blot pictures shown in Figure s2g is high. There are three bands appeared at d7, which one is for FBXO32?

RESPONSES TO REVIEWERS

Reviewer 1 (Remarks to the Author):

We thank the reviewer for many positive remarks and for the opinion that, ***“In the current manuscript of Sahu et al. the authors screen for novel mediators of EMT and identified FBXO32, which is required for TGFbeta mediated induction of EMT. The analysis shows that FBXO32 primarily interacts and ubiquitinates CtBP1 and that ubiquitination of CtBP1 alters its function and localization. CTBP1 ubiquitination is required to induce target gene expression in cooperation with ATF3. Analysis of human cancer specimen revealed that FBXO32 expression is increased in tumors and that high levels correlates with poor prognosis. Moreover, knockdown of FBXO32 in aggressive breast cancer cells decelerated tumor growth upon orthotopic transplantation and reduced metastatic burden. The authors conclude that FBXO32 is essential for EMT during development and disease.”*** He/she further comments that ***“The role of the EMT process in tumorigenesis and metastasis was recently challenged (Zheng et al., Nature 2015; Fischer et al., 2015), indicating that our understanding about the underlying mechanism of EMT and malignancy is still limited and views are controversial. Moreover, the function of posttranscriptional modifications in EMT was greatly underestimated and even less analyzed. Therefore, Sahu and colleagues are showing a novel and important mechanism about how EMT inducing signals can be processed and executed within the cell.”*** He/she continues with excitement for our study ***“The experiments are well designed, performed and analyzed and the manuscript is written in a concise way.”***

In addition the reviewer had few concerns that he/she suggested to be addressed before publication. We have addressed all points as described below:

Comment 1) A general conceptual problem is that the authors claim that they observe a reversion of the EMT phenotype when for example FXBO32 is knocked down. However, siRNA transfection is performed simultaneously with TGFbeta treatment. So very likely, the cells are blocked in an epithelial state and do not revert to it (E.g. on page 7, lines 137, 138, and following pages). This holds true for the differentially regulated genes on page 8: The term “up- and downregulated genes” is not precise enough. In fact, these genes are likely prevented from being regulated by TGFbeta and cells remain in an epithelial state. For the same reason the expression “mesenchymal cells depleted of FBXO32” on page 8, line 170 is also misleading, as these cells do not become mesenchymal.

Authors’ response: We agree with the reviewer that under our experimental conditions FBXO32 depleted cells are very likely to be blocked in an epithelial state and never undergo differentiation to a mesenchymal identity. To clarify this for the reader, we have now corrected according parts of the manuscript including the once mentioned by the reviewer. We thank the reviewer for pointing this out as this will facilitate the conceptual gain from our study.

Comment 2) Figs. S2g, Fig. 6b: With the exception of these two figures, the function of FXBO32 is mainly concluded from its differential regulation on mRNA level. It would be very important to demonstrate endogenous expression on protein level by WB or IF as well, especially its upregulation during EMT and reduction during siRNA knockdown (NMuMG, MDA-MB-231). Refs 31, 32 used antibodies that worked in WB and IHC, e.g. ab74023.

Authors' response: We fully agree with the reviewer that the endogenous expression of FBXO32 should be measured at the protein level too. We thank the reviewer for pointing out the references with the antibody information used in those previous studies. Unfortunately however, this antibody showed very high non-specificity under all of our tested conditions and cell types. A careful look at the abcam datasheet for this antibody confirms that it indeed detects many other non-specific bands (<http://www.abcam.com/fbx32-antibody-ab74023.html>). To solve this issue, we next tested a range of other available antibodies and following satisfactory performance of a selected set of antibodies for both human and mouse cells, we were able to analyze the endogenous FBXO32 protein levels under the following conditions:

- During EMT time course in mouse mammary epithelial (NMuMG) cells (new figure: Fig 1d)
- During EMT time course in human mammary epithelial cells, HMEC and HMLE (Old Fig 1e, new figure: Fig S5k)
- Following inhibition of protein translation during EMT in primary human mammary epithelial cells (HMEC) (new figure: Fig 3e)
- Following siRNA mediated depletion of FBXO32 during EMT in mouse mammary epithelial (NMuMG) cells (new figure: Fig S2h)
- Following siRNA mediated depletion of FBXO32 during EMT in primary human mammary epithelial cells (HMEC) (new figure: Fig S3a)
- Following siRNA and shRNA mediated depletion of FBXO32 in breast cancer cell line (MDA-MB-231) (new figure: Fig S11e)

Importantly, all these data are fully consistent and supportive of our claims made previously using Fbxo32 mRNA levels. To summarize the key observations from these new experiments, we show an increased FBXO32 expression at both mRNA and proteins levels during TGF- β induced EMT in mouse and human mammary epithelial cells. Moreover, siRNA-mediated depletion of FBXO32 with pooled or individual siRNAs during EMT in mouse and human mammary epithelial cells led to a strong reduction in FBXO32 expression at both mRNA and protein levels. A similar observation was made in breast cancer cells following siRNA and shRNA mediated depletion of FBXO32. We thank the reviewer for suggesting these experiments as these have helped significantly strengthen our observations.

Comment 3) The analysis on K63/K48 ubiquitination that results in the conclusion that FBXO32 induces CtBP1 ubiquitination via K63 is not convincing! This needs to be supported by additional experiments, e.g. by including other known substrates of FBXO32, inhibition of ubiquitination (inhibitor UBEI-41/PYR-41) or using a non-ubiquitinatable version of CtBP1. Alternatively, the statements in the results, discussion and especially in the summary of the introduction sections needs to be weakened. In Fig. 3c, g it is not clear which of the bands represents CtBP1. Does it overlap with the size of the band in the myc-CtBP1 blot? The authors should include positions of molecular weight standards to all blots.

Authors' response: We apologize for not having provided fully convincing evidence for FBXO32 inducing CtBP1 ubiquitination via K63 residue and thank the reviewer for suggesting highly relevant experiments. As advised by the reviewer, we have now chemically inhibited ubiquitination using UBEI-41/PYR-41 to strengthen previously shown FBXO32-dependent ubiquitination effects. Interestingly, while CtBP1 showed an increase in the ubiquitination following FBXO32 overexpression, this effect was blocked upon treatment with UBEI-41/PYR-41 (Old figure: Fig 3d and new figure: Fig S5g).

To substantiate our observations, we have now also repeated the ubiquitination assay using specific mutants of ubiquitin (Ub-K48 to R and Ub-K63 to R). Interestingly, while K48 to R mutant ubiquitin did not show any obvious effect on CtBP1 ubiquitination, K63 to R mutant ubiquitin completely abolished the ubiquitination (new figure: Fig 3g). These results indicate that the K63 residue of ubiquitin is involved in the polyubiquitin chain formation on CtBP1.

We have now also indicated the position of molecular weight standards for all western blots. With respect to the reviewer's concern on which of the bands represent CtBP1 in previous figures Fig. 3c, g, we would like to clarify that in case of ubiquitin immunoblots, addition of each ubiquitin residue further adds approximately 8.5 kDa molecular mass and therefore a smear-like pattern is typically seen in western blots for a poly-ubiquitination state as in our case.

Comment 4) page 34, Fig. 4l: The identification of ATF3 as candidate to cooperates with CtBP1 for transcriptional activation is not convincing. In particular, for ChIP at least day 0 untreated control should be included in the TGFbeta experiment. In addition, siFXBO32 or siCtBP1 need to be done in this experiment. In general ChIP experiments should be consistently displayed as percentage of input and an IgG control sample included for some experiments, especially in Fig. 4l. In line with the conclusion about ATF3: Fig. S6b shows ATF3 binding at the FXBO32 in ChIP-Seq data sets. However, an ATF3 knockdown does not affect FXBO32 expression as shown in Fig. 4k. How is that explained? Presumably, ATF3-mediated gene regulation is either more complex or it is the wrong candidate. Is gene regulation by known CtBP1 protein complexes (with Snail, Zeb1/2) changing upon FXBO32 knockdown? Are E-boxes found in the analyzed promoter regions in downregulated gene promoters? Is it possible that FBXO32 is turning EMT-TFs Zeb1/2, Snail, Slug into activators?

Authors' response: We are thankful to the reviewer for these comments and for suggesting very useful experiments to strengthen our findings about ATF3 function at CtBP1 target sites during EMT. As advised, we have now included the IgG negative control of the same antibody species as well as plotted the enrichment as % of input in the ChIP assays (new figure: Fig 4k). These results clearly show high enrichment for ATF3 at target promoter sites while no such enrichment was observed for IgG control.

We also performed ATF3 ChIP experiments in untreated cells and its comparison with those in TGF- β treated cells showed that ATF3 occupies the tested sites even prior to TGF- β stimulation (new figure: Fig 4k). These results imply that ATF3, being a sequence-specific factor, is pre-bound to these sites prior to CtBP1 targeting during EMT. Interestingly however, depletion of either CtBP1 or FBXO32 in cells undergoing TGF- β induced EMT strongly reduces ATF3 enrichment at target sites, suggesting a more complex crosstalk between these factors during mesenchymal progression (new figure: Fig 4l).

Altogether, these findings suggest that while ATF3 can target CtBP1 target sites in untreated epithelial cells, the stability of its binding during EMT relies on an active FBXO32-CtBP1 axis. Future studies should disentangle such interesting ATF3 binding behavior at CtBP1 target sites during acquisition of a mesenchymal fate.

With regard to the reviewers concern on Fbxo32 expression following ATF3 knockdown, while the downregulation in Fbxo32 expression is statistically significant, we agree that the extent of change is not very pronounced. However, as reviewer comments, ATF3-mediated gene regulation during EMT seems to be a bit more complex. For example, upon Fbxo32 knockdown, all tested targets loose CtBP1 occupancy except Fbxo32, which is the only candidate that does not show strong transcriptional downregulation following ATF3 depletion (Old Figures: Fig 4d, S8b).

Further following on reviewer's comments, we next attempted to investigate whether the gene regulation by known CtBP1 protein complexes (with Snail, Zeb1/2) is altered upon FBXO32 knockdown. Here we scanned for Snail, Zeb1/2 binding motifs (derived from TRANSFAC database) at the promoters of genes downregulated upon FBXO32 depletion that showed no significant enrichment of these motifs at these sites. We infer this to arise from the less complexity of these motifs (i.e. information content) as defined by the existing databases. This may be partly contributed by lack of availability of many good quality ChIP-seq datasets for these factors. Furthermore, peaks from existing ENCODE ChIP-seq datasets for ZEB1 and SNAIL showed minimal overlap with the promoters of FBXO32 and CtBP1 downregulated genes, suggesting that the gene regulation by known CtBP1 protein complexes (with Snail, Zeb1/2) is only partly affected upon FBXO32 knockdown.

We also analyzed occurrence of E-boxes (as defined by TRANSFAC) in the promoter regions of genes downregulated upon FBXO32 depletion. These results show no significant enrichment of E-box motifs at these sites.

Comment 5) The authors should try to confirm the FBXO32/CTBP1 interaction by IP of endogenous proteins in NMuMG, HMEC or MDA cells, maybe in response to TGFbeta, when this interaction is induced. This would greatly improve the conclusion from the artificial overexpression experiments in 293 cells (do 293 cells respond to TGFbeta?).

Authors' response: We fully agree with the reviewer that we must confirm the FBXO32/CtBP1 interaction by IP of endogenous proteins. Towards this, we have now performed IP for endogenous FBXO32 and CtBP1 in MDA-MB-231 cells. These results show that the endogenous forms of these two proteins indeed interact with each other (new figure: Fig 3c). We thank the reviewer for this input as this has enhanced the validity of our claims of a complex formation between FBXO32 and CTBP1.

Comment 6) Fig. 3d: It is puzzling that CtBP1 levels are decreasing during TGFbeta treatment, as the authors claim that FBXO32 is upregulated upon TGFbeta mediated EMT (as observed in HMECs) and overexpression of FBXO32 is increasing CTBP1 levels. This should be discussed and can be supported by showing an FBXO32 WB!

Authors' response: We apologize for having confused the reviewer with this figure (Previous Fig 3d). This immunoblot was not to show CtBP1 dynamics during TGF- β induced EMT but to demonstrate its stability at various time points following translation blockage using cycloheximide. This immunoblot has now been further improved (new figure: Fig 3e) and we have now edited the text and figure legends to clarify our experiment and findings.

We have now also performed CtBP1 immunoblot during EMT in both human and mouse models, and the results show that its levels stay reasonably constant throughout the EMT process (new figures: Fig S5d-e). This finding is interesting in the context of our observations that the regulation of its nucleo-cytoplasmic shuttling may be of utmost importance in CtBP1-dependent gene regulation during EMT.

Comment 7) Fig. 6: As mentioned above, the in vivo analysis of FBXO32 in tumorigenesis could be more thorough: Valuable information can be extracted from sections of primary xenograft and metastatic lungs stained by H&E, for EMT markers and for some candidates like MMPs, HAS2, etc. qRT-PCR of same (as in HMECs) or new top candidates in MDA cells upon FBXO32 knockdown (including EMT markers) should be done. Are the same genes affected

in MDA-MB-231 FBXO32 knockdown cells as in the initial screen? Is CtBP localization changing in MDA-MB-231 shFBXO32 cells?

Authors' response: We thank the reviewer for suggesting to further explore our *in vivo* tumorigenicity and migration analysis in the mouse model. The MDA-MB-231 is a mesenchymal cell line and does not exhibit many similarities with the primary EMT model systems such as in primary human mammary epithelial cells (HMEC). This means that many events that contribute to progression and acquisition of a mesenchymal state (e.g. during TGF- β induced EMT in HMEC) may not be required for the maintenance of mesenchymal identity such as in these cell lines like MDA-MB-231. Our results suggest that FBXO32 plays a role in both the progression of EMT and migration of mesenchymal cells in HMEC cell, while it only contributes to migration in metastatic breast cancer cells MDA-MB-231. This is also in line with our observations that the genes downregulated upon FBXO32 depletion during EMT in HMEC and in MDA-MB-231 show very little overlap. Along these lines, genes such as HAS and most of the MMPs are not expressed in MDA-MB-231 cells, further suggesting that many of these molecules may be required for progression and acquisition of mesenchymal identity but not for its maintenance.

Given these aspects, following reviewer's suggestion, we now selected few migration relevant genes that were commonly downregulated following *FBXO32* depletion during primary EMT in HMEC cells and in MDA-MB-231 cells for further analysis in primary tumors from mouse xenograft experiment. This analysis showed that these genes were significantly downregulated in their expression in tumors depleted of *FBXO32* as compared to the control tumors (new figure: Fig 6m).

As suggested by the reviewer, we also performed H&E staining on these tumors which showed a more compact organization of cells in *FBXO32* depleted primary tumors as compared to the control tumors which may be linked to the reduced metastatic behavior of *FBXO32*-depleted tumor cells (new figure: Fig 6n).

We also monitored protein products of few key secretory cytokines such as IL8, NGFR and GM-CSF (encoded by genes downregulated following *FBXO32* depletion in MDA-MD-231 cells) by immunohistochemistry which showed clear downregulation in *FBXO32* depleted tumors (new figure: Fig 6o).

We have also analyzed for CtBP1 localization in MDA-MB-231 cells upon *FBXO32* depletion in culture as well as in tumors from xenograft experiments. While we had difficulty in detecting a clear change in nucleo-cytoplasmic distribution due to tissue architecture and cell type, we observed a strong decrease in the nuclear intensity for CtBP1 following *FBXO32* depletion (new figures: Fig S11h-i).

Comment 8). page 15, Fig 6c: Colony formation was analyzed here, but to show what? In the methods section a cell survival assay is described which is not a clonogenic colony forming assay. The way it was performed, it simply shows cell viability upon trypsinization. I suggest to include a more thorough analysis of proliferation and cell death, especially since it is known that *FBXO32* is regulating apoptosis as well and increased apoptosis might challenge the xenograft experiment. The results of this will also affect Figs 6d,e. Maybe cells do not survive during plating for migration/invasion and therefore fewer cells are found on the other side of the transwell filter for analysis in the *FBXO32* situation? Moreover, a real sphere forming assay in presence of methylcellulose and absence of serum would be very valuable as an *in vitro* tumorigenicity assay.

Authors' response: We fully agree that further analysis of proliferation and apoptosis following FBXO32 depletion would be necessary to support our observations. We have now analyzed the rate of proliferation and survival in control and FBXO32 depleted cells (both in siRNA and shRNA treated cells) and no significant differences were detected except that in shRNA depleted condition we observed even more number of cells (new figures: Fig S11c-d).

Following reviewer's suggestion, we have now also performed sphere-forming assay in the presence of methylcellulose and absence of serum. Analysis from this experiment showed no obvious differences in primary and secondary sphere size and morphology as well as in number of cells between control and FBXO32 depleted MDA-MB-231 cells (new figures: Fig S11f-g).

In addition, we have also analyzed cleaved-caspase levels, which only showed a marginal increase upon FBXO32 depletion suggesting no intense apoptotic activity in these cells (new figures: Fig S11e). Altogether, our observations conclude that no significant changes in cell proliferation and survival occurs following FBXO32 depletion and rule out any possible influence of apoptosis on our key observations. We thank the reviewer for suggesting these additional experiments that have helped strengthen our conclusions.

Comment 9). The gene set enrichment analysis in Fig. 5h,i, Fig. S7 is lacking NES, FDR and p-values to make the analysis valuable. Are these changes significant? How were the GSEAs generated? By multiple analysis of many genesets simultaneously (how many and which ones?) or by individual comparison, as this will affect the outcome of the analyses?

Authors' response: We thank the reviewer for pointing this out. We have now included significance measures for enrichment differences, i.e. p-value, ES, NES and FDR in the respective figures. The p-values here were calculated by Wilcoxon test to quantify significance of fold change difference in expression values of gene sets in the two categories (i.e. patients with high and low FBXO32 levels). These statistical analyses show that NES and p-value are significant for all shown cases; ES and FDR are significant in most cases while being borderline in few cases. For the GSEA analysis, we had used multiple (n=5) gene sets associated to top five gene ontology terms that were enriched for FBXO32 downregulated genes in order to see their enrichment in FBXO32 high (Quartile 1) or low (Quartile 4) patients. The Previous Supplementary figure 7-8 are now Supplementary figures 9-10. We thank the reviewer for pointing us to include these values in the manuscript, which will clearly improve reader's appreciation and significance of our findings.

Comment 10). The results of Fig. 5b-e, S7, S8 are somewhat controversial with e.g. Refs 31 and 32. In these publications authors describe that FXBO32 acts more like a tumor suppressor in gastric cardia adenocarcinoma and urothelial carcinoma. These discrepancies should be discussed, maybe in line with the dual role of TGFbeta signaling during tumor progression (early: inducer of differentiation -> tumor suppressor; late: inducer of EMT and stemness -> tumor promoter) and frequent loss of SMAD4 in many tumors.

Authors' response: We also notice the differences in findings from these publications with respect to our own observations made in multiple model systems including primary EMT, metastatic cancer cell lines and clinical samples. We thank the reviewer for encouraging us to consider the known dual role of TGF- β signaling during tumor progression to explain these differences. As advised, we have now used this as a basis to elaborate in discussion about the discrepancies in findings for the role of FBXO32 in different cellular and disease contexts.

Minor points:

Comment 1). Fig S9b: MDA cells are already very mesenchymal, why are they treated with TGFbeta? One would expect that FBXO3 knockdown would induce MET in steady state conditions without TGFbeta. Measuring differentially regulated genes in untreated MDA shFXBO32 cells would be helpful to dissect this.

Authors' response: We apologize for creating confusion but MDA-MB-231 cells were cultured without TGF- β . We have now clarified this in the figure legend. Furthermore, we regret to not have highlighted that we had performed and supplied RNA-seq following FBXO32 depletion in MDA-MB-231 cells (GEO GSE77408). In addition, many of the downregulated genes including key cytokine genes critical for migration were further validated by independent RT-qPCRs (Figure: Fig S11b).

Comment 2). Did the authors check whether Fbxo32 is also regulated during other TGFbeta expression profiles (e.g. by the Christofori lab?)

Authors' response: We have adapted the same system that Christofori lab has been using to study EMT. Analysis of unpublished microarray data from TGF- β induced EMT in NMuMG cells from Christofori lab confirmed Fbxo32 induction during EMT (personal communication). Importantly further, we have also performed time course RNA-seq on the same system (Sahu et al. EMBO J. 2015) and these data support a significant upregulation of Fbxo32 during TGF- β induced EMT (Figure S2c).

Comment 3). page 40, model Fig. 7: The epithelium looks weird, like it is stratified with multiple basement membranes.

Authors' response: We thank the reviewer for pointing this out. We have now corrected the model.

Comment 4). Fig. S2d: inhibitors are not specified in the manuscript methods

Authors' response: We have now provided details of the inhibitors in the manuscript.

Comment 5). IP and ubiquitination assays were not described in methods.

Authors' response: These assays are now described in the methods section including the new ones performed during the revision.

Comment 6). Fig. 6i: please include the day of analysis for metastasis

Authors' response: This detail has been now provided.

Comment 7). Fig. S5b is redundant and should replace Fig 4b

Authors' response: We thank the reviewer for pointing this out. We have now removed Figure S5b.

Comment 8). page 16, line 337: Figs. S9d,e are not present in the current version

Authors' response: We apologize for this mistake which is now corrected.

Comment 9). p. 9, line 292: "FBXO32 levels correlated positively with the levels of established mesenchymal markers and negatively with the levels of epithelial markers in different types of

tumor examined (Fig S7e-f).” Only CDH1 was shown as epithelial marker. Either others should be included or sentence rephrased.

Authors’ response: We had checked few other established epithelial markers in our analysis. Although we saw a trend of negative correlation for few other epithelial markers such as TJP1, CLDN3 with FBXO32 levels, but this was not significant in all tumor types examined. Therefore, as suggested by the reviewer, we have now rephrased the sentence accordingly.

Comment 10). page 18, line 372: “nude mouse xenograft”. In the methods and results it is stated that NSG mice were used.

Authors’ response: We apologize for this mistake; we have now corrected the text.

Comment 11). Typos: line 310: “NSG mice model” -> NSG mouse model; line 461: “A previously described, a subclone of...” -> as previously described; legend to Fig. 1: “(e-f) ...the levels of BXO32 and key” -> FBXO32; p. 34, Fig 4j: containingig.

Authors’ response: We apologize for these typos and thank the reviewer for the careful reading. We have now edited the manuscript and corrected typos.

Reviewer #2 (Remarks to the Author):

“In the manuscript (NCOMMS-16-29334) entitled “FBXO32 mediates transcriptional program critical to promote microenvironment underlying epithelial-mesenchymal transition”, the authors discovered that FBXO32, an E3 ubiquitin ligase, acts as an upstream regulator of epithelial-mesenchymal transition (EMT) by governing the gene expression, and might play important roles in cancer metastasis and brain development.” The reviewer further comments “This study provided an in-depth analysis of FBXO32 on its function and underlying mechanisms, which might contribute to the understanding of EMT during disease and development.” We thank the reviewer for these encouraging comments on our study. In addition the reviewer raised concerns as he/she says ***“The observation is interesting, but the authors did not provide solid evidences for the novel mechanism of FBXO32 in EMT, particularly for the functional relationship between FBXO32 and CTBP1. The study is premature, and the data cannot fully support the conclusions.***

We carefully went through all points raised by Reviewer 2 and addressed each of them as follows:

Comment 1). The decrease of some of EMT marker genes, such as N-cadherin, Phalloidin and Paxillin, after FBXO32 depletion (shown in Figures 1D and s2i) is not that severe as authors claimed in the text, and these EMT markers are better organized in categories like epithelial or mesenchymal for clarity. How about their mRNA/protein levels?

Authors’ response: We apologize for the inconvenience caused to the reviewer that we believe arose due to reduction in the image resolution and size. This was required to allow successful upload of the originally very large immunofluorescence image files at the journal’s website. Provided our manuscript is accepted for publication, we will request the Editor to publish a higher resolution version of our images, as this is very much required to appreciate our findings. For now, we have placed all the original high-resolution images in a Dropbox folder for your consideration, which can be accessed here: <https://www.dropbox.com/sh/wjm6nnmfqkmc15c/AACHbrtnsIKmWU8Hen5bLVska?dl=0>

Following reviewer’s suggestion, we have now also organized the figure panels as epithelial markers (E-Cadherin, ZO-1), cytoskeleton reorganization markers (Paxillin, Phalloidin) and mesenchymal markers (N-Cadherin, Fibronectin). As the reviewer may appreciate from these high quality images, all these show expected behavior during normal EMT and these changes are strongly blocked when Fbxo32 is depleted during the process (modified figures: Fig1f and S3b).

To further support the results of immunofluorescence experiments, we also provided the quantitative measurement of the RNA levels of key EMT markers following FBXO32 knockdown in both human and mouse model systems (figure: Fig1g-h). In addition to immunofluorescence and RT-qPCR assays, we now also show by immunoblot analysis that key EMT markers (CDH1, CDH2, FN1 and ZEB1) show noticeable reversion upon FBXO32 depletion during EMT (new figure: Fig S2h).

Altogether, these three independent assays clearly establish significant changes in EMT markers following FBXO32 depletion. We thank the reviewer for motivating us to perform these changes and additional experiments, which have helped increase the significance of our findings.

Comment 2). While most of the experiments were done by using siRNA knockdown, a rescue experiment is required to confirm the function of FBXO32 in EMT, i.e., use siRNA-resistant FBXO32.

Authors' response: Following reviewer's suggestion, we now also attempted to perform the rescue experiment during FBXO32 depletion. We found that a long-term overexpression of FBXO32 was toxic for the cells and an increase in cell death was observed. Therefore, we performed a short-term (two days) overexpression of siRNA-resistant FBXO32 during EMT with simultaneous siRNA-mediated depletion of Fbxo32 and analyzed cellular morphology and EMT markers by RT-qPCRs. These results show a clear reversal of cellular morphology in siRNA resistant FBXO32 expressing cells during Fbxo32 knockdown (new figure: Fig S2i). Interestingly, within this short period of experiment, the mesenchymal markers showed a strong reversal while the epithelial markers also begin to change (new figure: Fig S2j).

In addition to these rescue experiments, we had also confirmed majority of our key findings using independent siRNAs in independent cell types (previous figures: Fig1f-h, S3b-d). Furthermore, we have now provided immunoblots to confirm Fbxo32 downregulation at the protein level following these knockdown experiments (new figures: Fig3e, S2h, S3a, S5k).

Altogether, these results clearly establish the specificity of siRNAs employed in our study. We thank the reviewer for motivating us to perform these experiments that have strengthened our claims and increased the impact of our findings.

Comment 3). The author performed IP-MS to identify that CTBP1 as a partner protein with FBXO32, but the experimental design has not been described in detail. This experiment is better showed in TGF- β treated cells with endogenous FBXO32 pull-down. Additionally, all the Co-IP experiments were carried out using overexpressed plasmids, which cannot account for the endogenous interaction between FBXO32 and CTBP1 proteins during EMT process. A native Co-IP experiment should be provided.

Authors' response: We apologize for not having provided full experimental details for the IP-MS experiments, which we have now included in the revised version of the manuscript. We fully agree with the reviewer that we must confirm the FBXO32/CtBP1 interaction by IP of endogenous proteins. Considering all cell types used in our study, we found MDA-MB-231 cells to have the highest endogenous expression levels of both FBXO32 and CtBP1. We next performed IP for endogenous FBXO32 and CtBP1 in these cells and the results show that indeed the endogenous forms of these two proteins interact with each other (new figure: Fig 3c). We thank the reviewer for advising to perform these experiments.

Comment 4). The author found that FBXO32 affected the CTBP1 nuclear retention via K63-linked ubiquitination, instead of K48. Whether this K63-dependent ubiquitination is essential for FBXO32-mediated EMT and CTBP1 nuclear retention is not known. Thus, K63 site-mutation should be performed to confirm the direct functional relationship between K63-dependent ubiquitination and CTBP1 nuclear retention.

Authors' response: We thank the reviewer for this highly valuable suggestion. We have now repeated the ubiquitination assay using specific mutants of ubiquitin (Ub-K48 to R and Ub-K63 to R). We find that while K48 to R mutant ubiquitin did not show any obvious effects on CtBP1 ubiquitination, K63 to R mutant ubiquitin completely abolish the ubiquitination (new figure: Fig 3g).

In addition to this experiment suggested by the reviewer, we have now also chemically inhibited ubiquitination using UBEI-41/PYR-41 to strengthen previously shown FBXO32-dependent ubiquitination effects. Interestingly, while CtBP1 showed an increase in the ubiquitination following FBXO32 overexpression, this effect was blocked upon treatment with UBEI-41/PYR-41 (old figure: Fig 3d and new figure: Fig S5g). We thank the reviewer for motivating us to perform these important experiments, which have provided strong support to our claims.

Comment 5). In Figure 4, the authors showed that CTBP1 depletion shared many EMT related genes with FBXO32 knockdown. FBXO32 depletion results in a lower binding of CTBP1 on EMT gene promoters. The authors concluded that FBXO32 mediates transcriptional activation of key EMT genes via nuclear CTBP1 targeting to their promoters and epigenetic remodeling, but these data cannot fully support the conclusion. As we mentioned above that the functional role of CTBP1 nuclear retention in EMT has not been fully characterized, the author need to overexpress FBXO32 in CTBP1 depleted cells to examine how the EMT is affected (like in Figure 4c).

Authors' response: We agree with the reviewer and following his/her advice; we have now analyzed the effect of CtBP1 depletion in FBXO32 overexpressing HMLE cells. Considering the undesirable consequences of long-term overexpression of FBXO32 (as explained in response to comment 2), we performed these assays over a short-time period of two days. Gene expression analysis by RT-qPCR showed that most tested CtBP1 target genes show reversal in their expression upon CtBP1 knockdown in FBXO32 overexpressing cells (new figure: Fig S7c).

Comment 6). Did FBXO32 promote tumorigenicity and cancer metastasis (Figures 5 and 6) through CtBP1?

Authors' response: We conclude from our *in vivo* experiments that FBXO32 possesses both tumorigenic and metastasis-promoting function. This is further supported by the finding that FBXO32 levels are frequently elevated in human tumor datasets. Furthermore, we have provided clear mechanistic insights into its role in EMT, which involves CtBP1-dependent gene regulation by epigenetic remodeling. In our revised manuscript, we also show that FBXO32 and CtBP1 endogenously interact with each other in metastatic breast cancer cells (new figure: Fig3c), which were also used for *in vivo* tumorigenicity and metastasis assays. While the results of cell culture and NSG mouse experiments are in agreement with respect to FBXO32-CtBP1-axis-dependent cell migration and metastasis respectively, we did not exclude the possibility that CtBP1-independent effects of FBXO32 may also play a role in tumor growth and metastasis.

We have now gone back to the primary tumor samples derived from mouse xenograft experiments and performed RT-qPCR and immunoblot analysis, which showed a significant downregulation of secretory cytokines such as IL8, NGFR and GM-CSF in tumors depleted of FBXO32 as compared to the control tumors at both RNA and protein levels (new figures: Fig 6m and 6o). These genes were found to be commonly downregulated following depletion of FBXO32 and CtBP1 during EMT in mammary epithelial cells as well as following knockdown of FBXO32 in breast cancer cells (MDA-MB-231), both contexts where FBXO32 forms a complex with CtBP1 (old figures: Fig3a, 3c). In line with these observations, we observed a strong decrease in the nuclear intensity for CtBP1 following FBXO32 depletion in tumors derived from mouse xenograft experiments (new figures: Fig S11h-i).

We now included these new data in the manuscript as well as discuss the possible scenarios in detail in the Discussion section. We thank the reviewer for motivating us to discuss these details further, which will benefit reader's understanding.

Minor points:

Comment 1). The immunohistochemical pictures showing cells in Figures 1D and s2i are not convincing enough. Authors should represent Z-stack and/or 3D views.

Authors' response: We apologize for the low resolution images provided with the manuscript. As explained in response to comment 1, we have now provided high resolution, high quality images for all figures in a Dropbox folder which clearly show strong changes in the shown epithelial and mesenchymal markers. As explained in detail in comment 1, we have now also enhanced the quality of our images, reorganized the figure panels as well as provided measurements of EMT markers at the RNA and protein levels following FBXO32 knockdown in both human and mouse model systems which show a clear EMT reversal upon FBXO32 depletion (figures: Fig 1g-h, new figure: FigS2h).

Link to Dropbox for high resolution images:

<https://www.dropbox.com/sh/wjm6nnmfqkmc15c/AACHbrtnsIKmWU8Hen5bLVska?dl=0>

Comment 2). Please specify the bands for Control in Figures 2c and 2d.

Authors' response: These are the controls provided by the Human MMP and Cytokine array kit (abcam: ab134004, R&D Systems: ARY005B), which are supposed to act a loading controls in our analysis to allow comparison of the blots. Unfortunately, the actual identity of these proteins are not disclosed by the company, these are most likely biotin-conjugated IgG printed on array to recognize by Streptavidin-HRP. We have now specified this in both figure legends.

Comment 3). The background of Western-blot pictures shown in Figure s2g is high. There are three bands appeared at d7, which one is for FBXO32?

Authors' response: We had tested several antibodies and performed western blots with the most specific antibodies, which showed loss of the signal for FBXO32 at the right size (42 kDa) in the depletion experiments. We now show several new western blots performed in various mouse and human cell lines (NMuMG, HMEC, HMLE, MDA-MB-231) of which only HMEC and HMLE cells show the three-band-pattern. Based on the results from other western blots, the most intense band in HMEC (previous figure s2g, now figure 1e) is likely the correct FBXO32 band while the other two lower faint bands are potentially degraded products or variants of FBXO32 which are only expressed in these cells.

Reviewers' Comments:

Reviewer #1:

Remarks to the Author:

The authors appropriately addressed all my points. I recommend publication of this manuscript in Nature Communications. The additional experiments and text modifications substantially improved the manuscript. Congratulations! This is really great work!

Reviewer #2:

Remarks to the Author:

No further comment.

Reviewer #3:

Remarks to the Author:

Background:

CTBPs are co-regulator proteins that together with TFs (SNAIL, ZEB etc.) modulate gene expression. CTBPs form complexes with epigenetic regulators that recruit epigenetic regulators.

Summary:

FBXO32 is induced during EMT, and plays a role by governing the required gene expression program.

FbxO32 dependent K63-linked ubiquitination of CTBP1 is required for nuclear retention, needed for epigenetic reprogramming of EMT target genes (chemokines, receptors, metalloproteinases).

FbxO32 is amplified in cancers, and depletion impairs metastatic properties of cancer cells.

Major point 1:

Overall, experiments to show the specificity of the interaction are missing in both the overexpression IP experiments (3b) and the ubiquitination experiment (3d).

- Fbox-mutant of FBXO32
- CTBP1 binding-mutant

Minimally, the authors should show that an fbox-domain mutant of FBXO32 is not able to bind and ubiquitinate CTBP1.

Major point 2:

The authors state on page 11 'that FBXO32 directly interacts with and modifies CTBP1 to promote its stability and nuclear retention'. This is confusing, because this implies that stability of CTBP1 is controlled by SCF-FBXO32-mediated degradation, however, the authors do not study the mechanism of CTBP1 degradation.

Minor points:

The authors state on page 11, line 231-233: quote 'It is known that FBXO32 can perform both K63 and K48-linked ubiquitination, which results in proteasome-mediated degradation or nuclear localization of target proteins, respectively'. This is the other way around. Please change.

Figure-by-figure minor points:

Fig 3. FbxO32-mediated K63-linked polyubiquitination of CTBP1 is required for the nuclear retention of CTBP1.

a. MS, triplo, 4 days after EMT induction by TGF- β .

- What about the other FBXO32 interactors from the Mass spec? Could they have a function in EMT?

b. IP of over-expressed proteins: IP-Myc-CTBP, co-IP-FLAG-HA-FBXO32

c. Endogenous IP: IP-FBXO32, co-IP of CTBP1.

- Is the observed band specific? The IgG-IP shows a background band similar to CTBP1 signal.
- Could the authors show vice versa interaction?

d. Ubiquitination experiment in 293T cells transfected with HA-Ub, Myc-CTBP1 and FLAG-FBXO32.

Anti-MYC-IP, co-IP of HA-Ub

- It is not clear what should be seen from the input-Ubiquitin membrane: Is there less ubiquitination in the control sample?

e. CHX chase after EMT induction.

- CTBP1 levels are lower in knock-down cells: is this via the UPS?
- Both FBXO32 and CTBP1 seem unstable (slow loss in CHX)

f. FBXO32 depletion causes localization of CTBP1 to the cytoplasm.

g. ubiquitination experiment in 293T cells transfected with HA-Ub (mutants), Myc-CTBP1 and FLAG-FBXO32.

Anti-Myc-IP, co-IP of HA-Ub

FBXO32 ubiquitinates CTBP1 via K63, not K48-chains.

- control is missing: overexpressed FLAG-FBXO32 levels

S5. FBXO32 mediates stability of CTBP1

S5c. IP of overexpressed proteins: IP of anti-HA-FBXO32, co-IP of Myc-CTBP1

S5de. CTBP1 protein levels stay stable during EMT induction in two cell types.

S5f. CTBP1 mRNA levels (by RNA seq) remain stable in FBXO32 depleted cells after EMT induction.

- In the figure, the CTBP1 mRNA levels seem decreased after FBXO32 depletion. The authors should explain/change this statement in the manuscript.

S5g. ubiquitination experiment in 293T cells transfected with HA-Ub, FLAG-O32 and Myc-CTBP1. IP of CTBP1, co-IP of HA-Ub. Chemical ubi-inhibitor, inhibits HA-Ub signal.

- controls are missing: input of FBXO32 and control sample without O32 and CTBP1
- it seems that the levels of IP-Myc-CTBP1 are lower.

S5h. overexpression of FBXO32 causes levels of CTBP1 to increase.

- control is missing: FBXO32 overexpression levels

S5i. FBXO32 knockdown causes levels of CTBP1 to decrease.

- control is missing: FBXO32 knockdown levels

S5j. same as h and i, but in another cell line.

- control is missing: FBXO32 knockdown/ overexpression levels

S5k. FBXO32 protein levels after EMT induction

- what should be concluded from this experiment?
- Western blots of figure S5h,i,j appear very faint. Could the authors provide blots with a longer exposure time?

S6. FBXO32 mediated K63 ubiquitination leads to nuclear retention of CTBP1

S6b. ubiquitination experiment in 293T cells transfected with HA-Ub (mutants), FLAG-O32 and Myc-CTBP1. IP of Myc-CTBP1, co-IP of HA-Ub.

K63 is all K's are mutant except K63. K48 is all K's are mutant except K48.

- Alternatively, this experiment indicates that additional (non-K63) lysines are needed for full ubiquitination. The authors should explain/ discuss that in the manuscript.

The original 1st and 2nd reviewer were fully satisfied with our revised manuscript and recommended publication. However, the newly added 3rd reviewer had few concerns that he/she suggested to be addressed before publication. We have addressed all of these points as described below:

Comments from Reviewer #3:

Major point 1: Overall, experiments to show the specificity of the interaction are missing in both the overexpression IP experiments (3b) and the ubiquitination experiment (3d).

-Fbox-mutant of FBXO32

-CTBP1 binding-mutant

Minimally, the authors should show that an fbox-domain mutant of FBXO32 is not able to bind and ubiquitinate CTBP1.

Authors' response: We agree with the reviewer that we should additionally assess whether the F-box-domain mutant of FBXO32 loses its ability to bind CTBP1 and hence ubiquitinate it. Towards this, we have now generated the FBXO32 deletion construct lacking the annotated F-box domain (known to interact with its substrate) and used this to perform IP assays. These results show that while the full length FBXO32 strongly interacts with CTBP1, the F-box domain mutant completely abolishes this interaction (new figure: S5d).

Major point 2: The authors state on page 11 'that FBXO32 directly interacts with and modifies CTBP1 to promote its stability and nuclear retention'. This is confusing, because this implies that stability of CTBP1 is controlled by SCF-FBXO32-mediated degradation, however, the authors do not study the mechanism of CTBP1 degradation.

Authors' response: We apologize for having confused the reviewer. Our data suggest that FBXO32-mediated K63 linked poly-ubiquitination inhibits degradation of CTBP1 as well as promotes its nuclear retention. This is in line with previous studies where it was shown that while the K63-linked poly-ubiquitination results in the activation of signaling pathways, K48 linked ubiquitination leads to proteasome-mediated degradation (M Akutsu, I Dikic, A Bremm et al., 2016). Future work should involve a deeper investigation to decipher the molecular mechanisms and machinery underlying such regulation.

Minor points:

Comment 1) The authors state on page 11, line 231-233: quote 'It is known that FBXO32 can perform both K63 and K48-linked ubiquitination, which results in proteasome-mediated degradation or nuclear localization of target proteins, respectively'. This is the other way around. Please change.

Authors' response: We thank the reviewer for pointing out this error in the text. We have now corrected this in the revised version of the manuscript.

Figure-by-figure minor points:

Fig 3. FbxO32-mediated K63-linked polyubiquitination of CTBP1 is required for the nuclear retention of CTBP1.

Comment 2) a. MS, triplo, 4 days after EMT induction by TGF- β .

- What about the other FBXO32 interactors from the Mass spec? Could they have a function in EMT?

Authors' response: Given our findings that FBXO32 depletion during EMT led to massive changes in the transcriptome and FBXO32 itself does not directly regulate gene expression, we were looking for a downstream transcriptional effector of FBXO32 through which it may execute these effects. Among FBXO32 interactors we identified in the mass-spec analysis, we chose CTBP1 as it is a known transcriptional regulator and has an established function in EMT. The other strongly enriched interactors of FBXO32 in the mass-spec analysis were not transcriptional regulators, but offer interesting candidates for a future investigation into their potential role during EMT.

Comment 3) b. IP of over-expressed proteins: IP-Myc-CTBP, co-IP-FLAG-HA-FBXO32

Authors' response: There was no concern from the reviewer associated to this comment.

Comment 4) c. Endogenous IP: IP-FBXO32, co-IP of CTBP1.

- Is the observed band specific? The IgG-IP shows a background band similar to CTBP1 signal.

- Could the authors show vice versa interaction?

Authors' response: We tested many commercially available antibodies for both FBXO32 and CTBP1 to validate endogenous interaction and only a few antibodies successfully worked in our hands. In the specific co-IP experiment mentioned by the reviewer, we find a clear significant enrichment as compared to the IgG control. Importantly, these observations were further supported by co-IP assays following overexpression of their wild type forms as well as an F-Box domain deleted version of FBXO32 (Figure 3b, S5c-d).

Comment 5) d. Ubiquitination experiment in 293T cells transfected with HA-Ub, Myc-CTBP1 and FLAG-FBXO32. Anti-MYC-IP, co-IP of HA-Ub

- It is not clear what should be seen from the input-Ubiquitin membrane: Is there less ubiquitination in the control sample?

Authors' response: Here we show total Ubiquitin levels in input samples as a control, which showed overall similar levels in all conditions.

Comment 6) e. CHX chase after EMT induction.

-CTBP1 levels are lower in knock-down cells: is this via the UPS?

- Both FBXO32 and CTBP1 seem unstable (slow loss in CHX)

Authors' response: Our analysis showed that CTBP1 is ubiquitinated by FBXO32 and inhibiting ubiquitination pathway results in a decrease in ubiquitination levels of CTBP1 (Figure S5h). Further analysis revealed that the depletion of FBXO32 leads to a decrease in the levels of CTBP1 (Figure 3e). This indirectly indicates that the lower CTBP1 levels in FBXO32 knockdown cells are via the UPS pathway.

With regard to the second point, majority of proteins in mammalian cells are expected to show a decrease in their levels following translational blockage (e.g. via CHX) with the exception of few highly stable proteins. This behavior is also reflected for FBXO32 and CTBP1 in our experiments.

Comment 7) f. FBXO32 depletion causes localization of CTBP1 to the cytoplasm.

Authors' response: There was no concern associated to this comment from the reviewer.

Comment 8) g. ubiquitination experiment in 293T cells transfected with HA-Ub (mutants), Myc-CTBP1 and FLAG-FBXO32. Anti-Myc-IP, co-IP of HA-Ub FBXO32 ubiquitinates CTBP1 via K63, not K48-chains.

- control is missing: overexpressed FLAG-FBXO32 levels

Authors' response: We have now added the control blot for FBXO32 that showed an expected profile.

Comment 9) S5. FBXO32 mediates stability of CTBP1 .

Authors' response: There was no concern associated to this comment from the reviewer.

Comment 10) S5c. IP of overexpressed proteins: IP of anti-HA-FBXO32, co-IP of Myc-CTBP1.

Authors' response: There was no concern associated to this comment from the reviewer.

Comment 11) S5de. CTBP1 protein levels stay stable during EMT induction in two cell types. (*New Figure S5e-f*)

Authors' response: There was no concern associated to this comment from the reviewer.

Comment 12) S5f. CTBP1 mRNA levels (by RNA seq) remain stable in FBXO32 depleted cells after EMT induction. (*New Figure S5g*)

- In the figure, the CTBP1 mRNA levels seem decreased after FBXO32 depletion. The authors should explain/change this statement in the manuscript.

Authors' response: There is indeed a slight tendency of downregulation in CTBP1 mRNA levels upon FBXO32 depletion. However, this change was very minimal (0.82 fold down i.e. less than 20% decrease) and does not appear significant to provoke any meaningful impact.

Comment 13) S5g. ubiquitination experiment in 293T cells transfected with HA-Ub, FLAG-O32 and Myc-CTBP1. IP of CTBP1, co-IP of HA-Ub. Chemical ubi-inhibitor, inhibits HA-Ub signal. (New Figure S5h)

- controls are missing: input of FBXO32 and control sample without O32 and CTBP1
- it seems that the levels of IP-Myc-CTBP1 are lower.

Authors' response: We have now repeated the experiment with additional controls (without FBXO32 only and without FBXO32 and CTBP1) and our conclusions remain unchanged. Moreover, these results also clearly show an enhanced ubiquitination following FBXO32 overexpression, which was diminished upon blockage of ubiquitin pathway by the addition of UBEI-41.

Comment 14) S5h. overexpression of FBXO32 causes levels of CTBP1 to increase. (New Figure S5i)

- control is missing: FBXO32 overexpression levels

Authors' response: We have now added the control blot for FBXO32 that confirmed overexpression of FBXO32 at the protein level.

Comment 15) S5i. FBXO32 knockdown causes levels of CTBP1 to decrease. (New Figure S5j)

- control is missing: FBXO32 knockdown levels

Authors' response: We have now added the control blot for FBXO32 that confirmed knockdown of FBXO32 at the protein level.

Comment 16) S5j. same as h and i, but in another cell line. (New Figure S5k)

- control is missing: FBXO32 knockdown/ overexpression levels

Authors' response: We have now added the control blot for FBXO32 that confirmed knockdown of FBXO32 at the protein level.

Comment 17) S5k. FBXO32 protein levels after EMT induction. (New Figure S5l)

- what should be concluded from this experiment?

Authors' response: Here we are showing induction of FBXO32 during TGF- β induced EMT in HMLE cells. We have used HMLE cells as an alternative of HMEC cells for biochemical and molecular analysis of FBXO32-dependent EMT regulation. Both model systems are very much accepted as complementary models in the field of EMT research.

Comment 18) - Western blots of figure S5h,i,j appear very faint. Could the authors provide blots with a longer exposure time? (New Figure S5i, j, k)

Authors' response: We have now added the longer exposure images for above mentioned blots.

Comment 19) S6. FBXO32 mediated K63 ubiquitination leads to nuclear retention of CTBP1

S6b. ubiquitination experiment in 293T cells transfected with HA-Ub (mutants), FLAG-O32 and Myc-CTBP1. IP of Myc-CTBP1, co-IP of HA-Ub. K63 is all K's are mutant except K63. K48 is all K's are mutant except K48.

- Alternatively, this experiment indicates that additional (non-K63) lysines are needed for full ubiquitination. The authors should explain/ discuss that in the manuscript.

Authors' response: We thank the reviewer for pointing out this critical observation. We have now discussed this in the results section of our revised manuscript.

Reviewers' Comments:

Reviewer #3:

Remarks to the Author:

Major point 1.

The authors did not eliminate the concerns regarding the specificity of the interaction. On the contrary, they raised additional confusion. In fact, they misinterpret how the F-box protein in the SCF complex determines substrate specificity: how it recognizes, binds and ubiquitinates its target substrates. For clarity, the F-box domain in the F-box protein binds Skp1, the adaptor protein interacting with the N-terminus of Cul1, which bridges to the catalytic subunit Rbx1-Ubc3. Hence, an F-box domain mutant of FBXO32 is predicted to lose interaction with the SCF-complex, not with the substrates. In contrast, F-box domain mutants of F-box proteins, often bind substrates more stably, because they work as substrate traps. F-box proteins bind their substrates through a degron, which could be identified by mapping the interaction domains of substrates for F-box proteins. The authors should reinterpret and explain their results of new figure S5d accordingly.

Reply to comment 4)

This is not very convincing, for three reasons:

- 1, The figure shows enrichment of CTBP1 signal in the FBXO32-IP compared to IgG, however, there is no indication that the same amount of antibody was used.
- 2, The authors refer to Fig 3b, where they show overexpressed protein-interactions. This would be a convincing addition, however, the FBXO32 blot shows an uneven background signal.
- 3, See above for comment on the f-box-domain mutants

Reply to comment 12)

The authors claim that "the slight tendency of downregulation does not appear to provoke any meaningful impact". What is this based on? And what do the error bars (that clearly do not overlap) indicate?

Reply to comment 13)

The authors claim that they repeated the experiment, however, the two bands of the CTBP1 blot in the old Fig S5g look identical to the two bands on the right on the CTBP1 blot of new Fig S5h.

Reviewer 3 had few more minor concerns which we have addressed as follows:

Comments from Reviewer #3:

Major point 1. The authors did not eliminate the concerns regarding the specificity of the interaction. On the contrary, they raised additional confusion. In fact, they misinterpret how the F-box protein in the SCF complex determines substrate specificity: how it recognizes, binds and ubiquitinates its target substrates. For clarity, the F-box domain in the F-box protein binds Skp1, the adaptor protein interacting with the N-terminus of Cul1, which bridges to the catalytic subunit Rbx1-Ubc3. Hence, an F-box domain mutant of FBXO32 is predicted to lose interaction with the SCF-complex, not with the substrates. In contrast, F-box domain mutants of F-box proteins, often bind substrates more stably, because they work as substrate traps. F-box proteins bind their substrates through a degron, which could be identified by mapping the interaction domains of substrates for F-box proteins. The authors should reinterpret and explain their results of new figure S5d accordingly.

Authors' response: We thank the reviewer for these details to help us better explain our data. To facilitate understanding of our findings, we have now added further details in the supplementary figure 4d (previously supplementary figure 5d) and the corresponding figure legend. This includes an schematic representation of full length human FBXO32 and its mutant version lacking the entire F-box domain. This deleted large F-box domain (115 aa) was predicted by InterPro which encompasses the core F-box domain (45 aa). Given these details, it is possible that the amino acids that were deleted in addition to the core F-box play an essential role in the interaction of FBXO32 with its substrate CTBP1. It is also likely that such deletion affects binding of other factors directly or indirectly which in turn results in a loss of FBXO32 interaction with CTBP1. Future work should involve a more fine mapping of amino acid residues that are involved in FBXO32 function including its substrate recognition.

Minor points:

Reply to Comment 4) This is not very convincing, for three reasons:

- 1, The figure shows enrichment of CTBP1 signal in the FBXO32-IP compared to IgG, however, there is no indication that the same amount of antibody was used.
- 2, The authors refer to Fig 3b, where they show overexpressed protein-interactions. This would be a convincing addition, however, the FBXO32 blot shows an uneven background signal.
- 3, See above for comment on the f-box-domain mutants

Authors' response: With regard to point 1, as a general practice in the field, we used exactly the same amount of lysate for each immunoprecipitation (IP) experiment with equal amount of antibody for IgG and bait proteins (in our case Flag antibody for FBXO32 and Myc

antibody for CTBP1). Furthermore, following IP experiments, they were eluted in an equal volume of elution buffer of which we loaded equal amounts in western blots for a comparison. We have now further expanded these details in our material and methods section. With regard to point 2, we have performed this experiment several times and obtained very similar results and thus we are convinced of our findings, which are further supported by a number of additional experiments presented throughout the manuscript. For point 3, please refer to our response to major point 1.

Reply to Comment 12) The authors claim that “the slight tendency of downregulation does not appear to provoke any meaningful impact”. What is this based on? And what do the error bars (that clearly do not overlap) indicate?

Authors’ response: Here we had plotted the normalized tag counts obtained from transcriptome analysis (RNA-seq) of biological replicates from control versus FBXO32 depleted HMEC cells during EMT. We agree that the error bars are very small and do not overlap in this bar plot, which reflect that the changes in the tag counts for control and FBXO32 knockdown conditions are highly reproducible within the biological replicates. However, the DESeq program (one of the most accepted computational tool to reveal statistically significantly differentially expressed genes between given conditions) used in this study showed that these changes are statistically not significant. For further details on this program, please refer to: <https://genomebiology.biomedcentral.com/articles/10.1186/gb-2010-11-10-r106>.

Reply to Comment 13) The authors claim that they repeated the experiment, however, the two bands of the CTBP1 blot in the old Fig S5g look identical to the two bands on the right on the CTBP1 blot of new Fig S5h.

Authors’ response: We apologize to the reviewer for this error and thank for pointing this out. As noticed from the upper part of this figure we had indeed repeated this experiment, but missed to replace the control below with the new one. The updated figure is now provided as supplementary figure 4h (previously supplementary figure 5h) in the revised manuscript.